# Towards W2A4 LLM Inference: Hybrid SQ-VQ Framework with Adaptive Error Compensation

## Abstract

Quantization presents a powerful approach for reducing the memory footprint and accelerating the inference of Large Language Models (LLMs). However, it faces a fundamental dilemma: computation-friendly Scalar Quantization (SQ) suffers performance degradation at ultra-low bit-widths, whereas memory-friendly Vector Quantization (VQ) maintains higher accuracy but fails to reduce computational demand. As a result, achieving both computational efficiency and high-fidelity compression in ultra-low-bit regimes (e.g.W2A4) remains a tough challenge. To address this, we propose **AEC-SVQ**, a hybrid framework that synergistically integrates SQ ,VQ for high-performance, ultra-low-bit LLM inference. The framework is built on three innonvations. ❶To simultaneously address the disparate distributional challenges presented by weight VQ, activation SQ, and codebook integer quantization, we introduce a **learned rotation-smooth transformation** that adaptively promotes quantization-friendly distributions for weights, activations, and codebooks within the hybrid SQ–VQ scheme. ❷To mitigate the compounding errors caused by the independent quantization of weights and activations, we propose the **Cumulative-Error-Aware Vector Quantization (CEAVQ) algorithm**. CEAVQ adjusts weights to compensate for the cumulative error from upstream quantized layers, thereby proactively aligning with the full-precision output distribution. ❸To ensure robustness against statistical noise from limited calibration data, we introduce a closed-form, data-driven **Adaptive Compensation**. It modulates the compensation strength for cumulative errors, preventing overfitting to calibration set statistics and guaranteeing stable generalization. AEC-SVQ enables a W2A4 pipeline that achieves the memory footprint of a 2-bit model while exploiting the computational efficiency of 4-bit integer arithmetic. On LLaMA-30B, it delivers a $3.6\times$ speedup and $7.1\times$ memory saving, establishing a practical frontier for ultra-low-bit LLM deployment.

## 1 Introduction

Large Language Models (LLMs) (Dettmers et al., 2022a; Touvron et al., 2023a;b) have unlocked remarkable capabilities across diverse domains (Achiam et al., 2023; Chen et al., 2024), yet their immense computational and memory footprints present a significant barrier to widespread deployment. The prohibitive cost of serving these models, particularly on resource-constrained edge devices, has catalyzed intensive research into model compression. Among various techniques, quantization—reducing the numerical precision of weights and activations to lower bits—stands out as one of the most promising avenues for dramatically cutting memory usage, bandwidth, and energy consumption.

As shown in Figure 1(c), LLM quantization is primarily driven by two approaches: Scalar Quantization (SQ) Ashkboos et al. (2024) and Vector Quantization (VQ) (Liu et al., 2024a). SQ, particularly in INT8 and INT4 settings, has gained wide adoption due to its seamless compatibility with commodity hardware, which offers highly optimized integer arithmetic pipelines for efficient computation. However, at sub-4-bit precision, the limited representational capacity of SQ causes severe accuracy loss. In contrast, VQ demonstrates distinct advantages in the ultra-low bit regime($< 4$ bits). By mapping weight parameters to high-dimensional floating-point (FP) codewords, it preserves key information while further improving the compression ratio. Despite its efficacy in reducing memory and bandwidth, current VQ methods are restricted to weight-only quantization and remain in costly FP arithmetic, stemming from two core issues: the prohibitive complexity of quantizing runtime

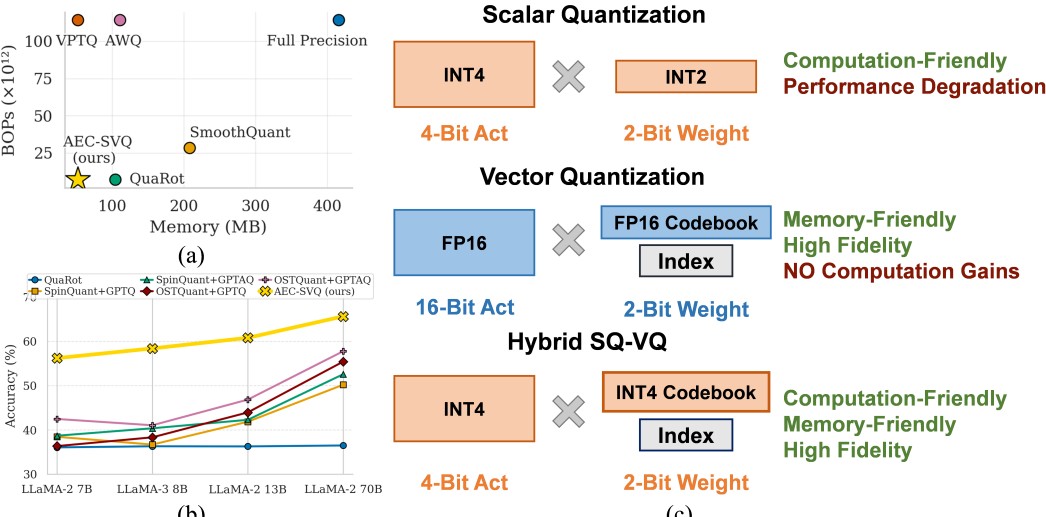

Figure 1: **Motivation and performance overview of our proposed hybrid W2A4 quantization framework, AEC-SVQ. (a)** Comparison of Bit Operations (BOPs) and memory footprint for recent quantization methods, demonstrating that AEC-SVQ achieves a superior integration of computation and memory efficiency. **(b)** Accuracy of various methods on the LLaMA family, where AEC-SVQ consistently outperforms existing PTQ techniques. **(c)** Conceptual comparison of quantization schemes. Our hybrid approach synergizes SQ and VQ, using an INT4 codebook to enable memory-friendly 2-bit weight storage while executing computation with efficient 4-bit integer arithmetic.

activations online and the incompatibility of its non-linear output with accelerated integer arithmetic. Taken together, the characteristics of VQ and SQ reveal a fundamental computation–memory trade-off and raise a key challenge: **How can we synergistically integrate the hardware-friendly efficiency of SQ with the high-fidelity representation of VQ into a unified framework to enable practical, high-performance ultra-low-bit inference?**

Building upon the characteristics of common hardware architectures and the requirements of LLM inference, we first explore a hybrid quantization scheme. This initial approach, which employs SQ for runtime activations and VQ for model weights, offers a promising balance of computational efficiency and compression. To align with low-precision compute units such as INT4 Tensor Cores and achieve practical speedups, the codebook in VQ are also quantized using SQ. However, the practical implementation of this strategy faces several critical challenges: ❶ **Suboptimal data distributions for quantization.** The intrinsic data distributions in LLMs are challenging for standard quantization methods. For weights, VQ performs best on isotropic, spherical clusters (Yue et al., 2025), but the typically anisotropic nature of weight distributions often leads to suboptimal representations. For activations, scalar quantization requires a narrow dynamic range to maintain high fidelity (Dettmers et al., 2022b). Yet, outliers in LLM activations drastically widen this range, forcing most values into coarse, low-information bins. For VQ codebook, the few codewords representing high-magnitude outliers skew the dynamic range, forcing subsequent SQ to collapse most other codewords into coarse bins and degrade overall fidelity. ❷ **Coupled quantization errors**. Conventional approaches quantize weights and activations independently, overlooking critical error interactions within and across layers. Distortions from weight quantization can shift data distributions, amplifying activation errors. Conversely, activation quantization alters the input statistics of downstream layers, rendering pre-calibrated weight reconstructions suboptimal. Under ultra-low bit widths, these uncompensated and accumulated errors become the main bottleneck, making simple independent schemes ineffective. A promising solution is to explicitly model and correct the coupled distortions and cumulative error. However, in contrast to independent optimization, holistic modeling and collaborative optimization are often compromised by ❸ **Statistical instability**. PTQ relies on small calibration datasets to estimate correction statistics. Limited sample sizes inevitably introduce noise into these estimates. Applying such noisy corrections indiscriminately can destabilize inference and degrade generalization. Therefore, a central challenge for practical deployment is to design a robust method that leverages beneficial corrections while mitigating the impact of statistical noise.

In response, we propose **AEC-SVQ** a Hybrid SQ-VQ framework to unlock an efficient W2A4 (2-bit weight, 4-bit activation) pipeline. We first construct a **Hybrid SQ–VQ scheme with Learned Transformation** designed to correct suboptimal data distributions for quantization, thereby making them more amenable to subsequent quantization. The effectiveness of this transformation is

validated through both theoretical derivation(Equation 4) and intuitive illustration(Figure 2(b, c)). By simultaneously satisfying the distributional requirements of activation SQ, weight VQ, and codebook quantization, the transformation reduces overall quantization error and integrates these three components into a unified optimization process, ultimately realizing a cohesive and efficient hybrid scheme. Building on this foundation, we tackle the coupled distortions and cumulative errors of independent quantization with our **Cumulative-Error-Aware Vector Quantization (CEAVQ).** Unlike conventional approaches that treat layers in isolation and minimize local reconstruction error, CEAVQ pursues a global objective: aligning each layer's output with its full-precision distribution. To this end, it introduces a novel corrective term that proactively adjusts the weights. This adjustment compensates not only for the imminent activation quantization error but also for the accumulated distortions propagated from upstream quantized layers. Finally, to address the practical issue of statistical instability, we propose **Adaptive Compensation via Bias–Variance Shrinkage**, which formalize the problem by modeling the application of the corrective term as a classical bias–variance trade-off. Based on this formulation, we derive a closed-form, data-driven solution that adaptively adjusts the strength of cumulative error correction at the granularity of individual columns. Acting as a theoretically grounded shrinkage mechanism, it suppresses unreliable correction signals and ensures stable, generalizable performance.

Specifically, our work makes the following three core contributions:

- **Hybrid SQ-VQ scheme with Learned Transformation**. We introduce a hybrid scheme built around a learned transformation that reshapes data distributions into quantization-friendly forms. This transformation simultaneously meets the distributional requirements of activation SQ, weight VQ, and codebook quantization, reducing overall quantization error and integrating the three components into a unified optimization process. As a result, the scheme enables a synergistic hybrid scheme that achieves the **memory footprint of a 2-bit model** while **retaining the computational efficiency of 4-bit integer arithmetic.**

- **Cumulative-Error-Aware VQ (CEAVQ) algorithm.** We propose a post-training quantization algorithm that aligns each layer's output with its full-precision reference. CEAVQ introduces a corrective term that proactively adjusts the weight vectors, compensating for both activation quantization errors and the accumulated distortions propagated from upstream layers. This coordinated optimization alleviates the compounding effects of independent quantization and enables more accurate ultra-low-bit inference.

- **Adaptive Compensation via Bias-Variance Shrinkage.** To improve the robustness of CEAVQ under statistical noise from limited calibration sets, we formalize the instability as a bias–variance trade-off. From this formulation, we derive a closed-form, data-driven compensation method that applies shrinkage to the corrective term. This adaptive mechanism balances correction strength against estimation noise, ensuring stable and generalizable performance.

## 2  RELATED WORK AND BACKGROUND

**Scalar Quantization for LLM Compression.**    SQ converts weights and activations of pretrained neural networks from high precision (e.g., 16-bit floating point numbers) to lower precision (e.g., 4-bit integers). Given a weight $\mathbf{W}$, it is typically implemented with symmetric and uniform quantization as:

$$\mathrm{SQ}(\boldsymbol{W}) = \mathrm{clamp}(\lfloor \frac{\boldsymbol{W}}{\boldsymbol{s}} \rceil, -2^{b-1}, 2^{b-1} - 1), \quad \boldsymbol{s} = \frac{\max(|\boldsymbol{W}|)}{2^{b-1} - 1}, \tag{1}$$

where $s$ is the scale factor, $\lfloor \cdot \rceil$ denotes the rounding-to-nearest operator, $b$ is the quantization bit-width, and $\mathrm{clamp}$ is the clipping function. SQ remains the workhorse of LLM compression due to its compatibility with integer arithmetic on commodity accelerators. In the weight-only regime, methods such as GPTQ (Frantar et al., 2022) leverage second-order error models to minimize rounding errors, while AWQ (Lin et al., 2023) and OWQ (Lee et al., 2023) employ activation-aware scaling to protect salient weights from quantization loss. To obtain end-to-end speedups, recent work pushes SQ to both weights and activations, where the key difficulty is that activation outliers dominate the dynamic range and lead to insufficient precision representation for most data points. ZeroQuant (Yao et al., 2022) proposes fine-grained, hardware-friendly schemes; SmoothQuant (Xiao et al., 2022) shifts dynamic range from activations to weights via an equivalent rescaling; OmniQuant (Shao et al., 2023) further learns quantization and transformation parameters; and I-LLM (Hu et al., 2024) redesigns blocks and operators to enable fully integer inference. Orthogonal transforms have emerged as a

complementary strategy to make SQ viable at 4 bits: QuaRot (Ashkboos et al., 2024) uses random rotations to deconcentrate outliers, while SpinQuant (Liu et al., 2024b) learns rotations to adaptively regularize distributions. OSTQuant (Hu et al., 2025) unified learnable rotations and scaling, providing additional flexibility and consistently outperforming previous methods. FlatQuant (Sun et al., 2024) employed layer-wise learned online matrix transforms to improve quantized linears, at the cost of increased inference overhead.

**Vector Quantization for LLM Compression.** VQ has emerged as a powerful technique for achieving higher compression ratios. The core idea of VQ is to map a large set of vectors to a smaller, finite set of representative vectors—commonly referred to as a codebook. Each original vector is then represented simply by the index of its closest counterpart in the codebook, achieving compression by storing this compact index instead of the full-precision vector. exploiting inter-channel correlation to attain lower distortion at the same bit budget than scalar quantization. Given a weight $W$ with $m$ rows and $n$ columns to be quantized, VQ reshapes it into $W'$ with dimensions $(m * n/v, v)$. For each $v$-dimensional row vector, VQ replaces it with the $log_2 k$-bit index of the nearest vector from the codebook $C \in \mathbb{R}^{k \times v}$. The compression ratio of VQ is $(16mn)/(16kv + log_2 k * mn/v)$. Typically, the Euclidean distance (calculated by the Frobenius normalization $|| \cdot ||_F$) is taken to measure similarities. In this case, the quantization process can be expressed as:

$$\text{VQ}(W') = \{\underset{j \in k}{\text{argmin}} ||W'_{i,:} - C_{j,:}||_F \,|\, i = 1, ..., m * n/v\}. \tag{2}$$

The codebook $C$ has shape $(k, v)$, where each row vector represents a cluster center. Codebook design is central to VQ methods. QuIP# (Tseng et al., 2024), leverage structured, data-independent codebooks for extreme compression. In contrast, a more common approach is to learn data-dependent codebooks. VPTQ (Liu et al., 2024a) and GPTVQ (Van Baalen et al., 2024) optimize codebooks using clustering algorithms like K-Means and Expectation-Maximization, respectively, with AQLM (Egiazarian et al., 2024) further refining this via layer-wise training. To mitigate error accumulation, both VPTQ and AQLM incorporate residual quantization. Another line of work exploits the geometric properties of weight vectors to improve quantizability: PCDVQ (Yue et al., 2025) decouples vector magnitude and direction, while PVQ (van der Ouderaa et al., 2024) constrains codewords onto a sphere to better match the weights' natural distribution.

## 3 METHODOLOGY

### 3.1 HYBRID SQ-VQ SCHEME WITH LEARNED TRANSFORMATION

We introduce a hybrid quantization scheme that synergistically combines VQ for weights and SQ for activations. This approach addresses their disparate statistical properties, as SQ is well-suited for the dynamic distributions of activations, while VQ offers superior rate-distortion performance for static weight tensors. However, a naive implementation of this method is suboptimal due to a misalignment between the intrinsic data distributions in LLMs and the ideal operating conditions for each quantization scheme. The effectiveness of VQ is predicated on isotropic, spherical data clusters. This condition is violated by the anisotropic geometry of weight tensors, as visualized in Figure 2(c, left) leading to inefficient codebook representations. The fidelity of SQ depends on a minimal dynamic range. This is severely undermined by emergent outliers in activations(Figure 2(b, left)), which drastically expand the quantization range and compel the majority of values into coarse, low-information bins.

To reconcile these requirements, we introduce a Learnable Rotation-Smooth Transformation, an equivalent transformation pair that reshapes the distributions of both weights and activations to be simultaneously amenable to their respective quantization schemes. We theoretically prove that this single transformation systematically benefits not only the weight VQ and activation SQ but also the subsequent integer quantization of the VQ codebook itself. This unification enables a highly efficient inference pipeline. By quantizing the VQ codebook to 4-bit integers (INT4), we convert the primary matrix multiplication (MatMul) into an INT4 table lookup and multiply-accumulate operation. Consequently, our framework achieves the storage and bandwidth advantages of 2-bit weights (W2) while leveraging the computational speedups of 4-bit arithmetic (W4A4), a complete workflow visualized in Figure 2(a).

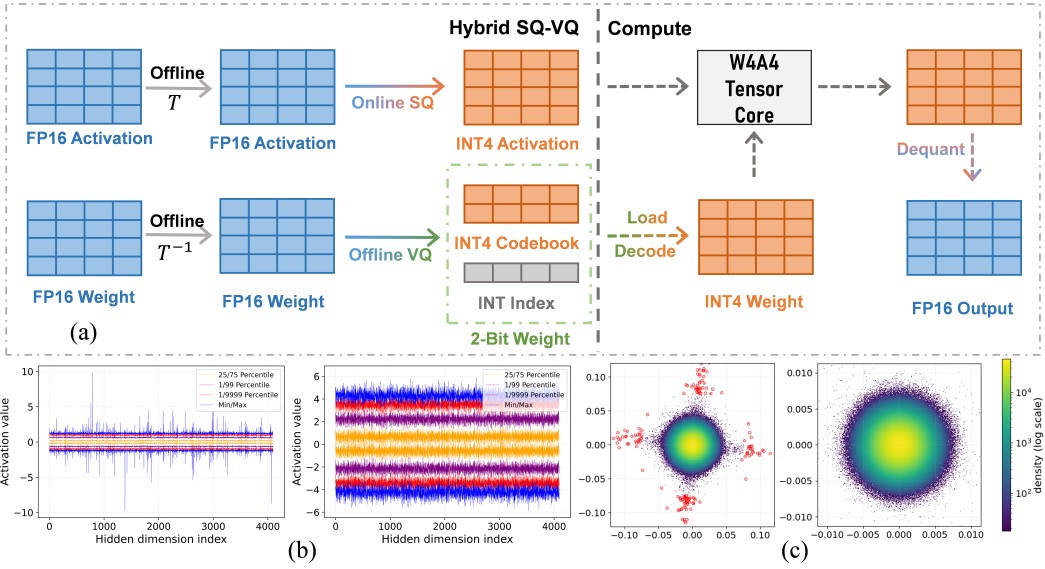

Figure 2: **Overview of Hybrid SQ-VQ scheme and the distributional effect of the learned transformation.** **(a)** W2A4 inference pipeline. Offline learned transformation pair$(\mathbf{T}, \mathbf{T}^{-1})$ reshapes data distribution. Activations subsequently undergo online 4-bit SQ, while weights are quantized offline using 2-bit VQ with an INT4 codebook. This scheme enables efficient computation on W4A4 tensor cores. **(b)** The transformation mitigates outliers in the raw activation distribution (left), producing a uniform and compact distribution (right) that is ideal for SQ. **(c)** PCA distribution of vectorized weights. The initially anisotropic weight distribution with outliers (left) is transformed into a dense, isotropic cluster (right), creating an optimal geometry for k-means-based VQ.

Consider a linear layer defined by $\mathbf{y} = \mathbf{W}\mathbf{x}$, where $\mathbf{x} \in \mathbb{R}^{d_{in}}$ is the input activation and $\mathbf{W} \in \mathbb{R}^{d_{out} \times d_{in}}$ is the weight matrix. We introduce an *equivalent transformation pair* $(\mathbf{T}, \mathbf{T}^{-1})$ that preserves the layer's function:

$$\mathbf{y} = \mathbf{W}\mathbf{x} = (\mathbf{W}\mathbf{T}^{-1})(\mathbf{T}\mathbf{x}) = \mathbf{W}'\mathbf{x}'. \tag{3}$$

The transformation $\mathbf{T}$ is parameterized as a learnable rotation-smooth operator, explicitly defined as $\mathbf{T} = \mathbf{\Lambda}\mathbf{O}$. Here, $\mathbf{O} \in \mathbb{R}^{d_{in} \times d_{in}}$ is a learnable orthogonal matrix (rotation) that mixes input channels, and $\mathbf{\Lambda}$ is a learnable diagonal matrix (smoothing) that adjusts the variance of each resulting channel.

Modeling the errors from activations SQ$(\eta_x)$, weights VQ$(\eta_w)$, and codebook integer quantization$(\eta_c)$ as additive noise. we derive a unified approximation for the layer's output Mean Squared Error (MSE):

$$\mathbb{E}\|\tilde{\mathbf{y}} - \mathbf{y}\|_2^2 \approx \underbrace{\text{tr}\left(W'^{\top}W'\mathbf{\Sigma}_{\boldsymbol{\eta}_x}\right)}_{\text{Activation Error}} + \underbrace{\text{tr}\left(\mathbf{x}'\mathbf{x}'^{\top}(\mathbf{\Sigma}_{\boldsymbol{\eta}_w} + \mathbf{\Sigma}_{\boldsymbol{\eta}_c})\right)}_{\text{Weight \& Codebook Error}} \tag{4}$$

, where $\Sigma$ represent the error covariances, This model reveals that the transformation $\mathbf{T}$ jointly influences all error sources by modifying both the activation statistics and the effective weight geometry.

Our core theoretical claim is that the proposed $\mathbf{T}$ systematically reduce all three constituent error terms in Equation 4, as empirically demonstrated in Figure 2(b, c). Theoretical analysis can be referred to Appendix A.2

### 3.2 CUMULATIVE-ERROR-AWARE VECTOR QUANTIZATION

Conventional PTQ methods that treat weight and activation quantization as independent, locally-optimized problems are fundamentally suboptimal. This approach overlooks the crucial coupling of their respective errors, which fosters a reciprocal error amplification Specifically, quantizing activations shifts the input statistics for weights, while weight quantization error perturbs the output, magnifying errors in subsequent layers. This cascade of uncompensated, compounding error, visualized in Figure 3 (bottom), becomes the primary performance bottleneck in ultra-low-bit regimes. An effective weight quantization strategy must therefore abandon this decoupled approach and instead actively compensate for the error induced by activation quantization.

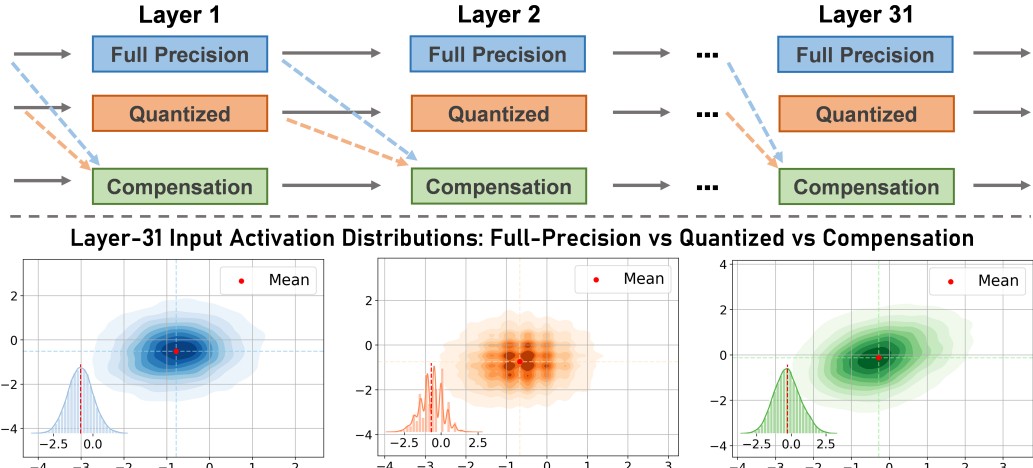

Figure 3: **Conceptual illustration of our Cumulative-Error-Aware Vector Quantization (CEAVQ). (Top)**A schematic of the layer-wise compensation mechanism. Error information from the quantization process in one layer is used to apply a corrective adjustment to the subsequent layer, proactively mitigating the accumulation of cascading errors. **(Bottom)**A comparison of the input activation distributions for Layer 31. The leftmost plot shows the ideal distribution under a full-precision model. The center plot reveals a significant distributional shift caused by standard quantization. The rightmost plot demonstrates that our CEAVQ method successfully preserves the statistical integrity of the original distribution, even deep within the network.

To this end, we propose a sequential, layer-wise compensation strategy, conceptually illustrated in Fgure3(top). We reformulate the weight quantization problem from a foundational perspective. Our goal is to find the quantized weight matrix $\hat{W}$ that minimizes the following principled objective:

$$\ell(\hat{W}) = \mathbb{E}_x \left[ \left\| \hat{W} X - W \tilde{X} \right\|_F^2 \right] = \mathbb{E}_x \left[ \left\| (\hat{W} - W) X - W(\tilde{X} - X) \right\|_F^2 \right] \tag{5}$$

where $X$ are the quantized activations corresponding to the full-precision inputs $\tilde{X}$.

As detailed in Appendix A.3, minimizing this objective reveals that the optimal, unquantized solution is not $W$, but rather an error-compensated matrix $W_{opt} = W + WGH^{-1}$. The term $WGH^{-1}$ serves as a corrective pre-shift to the weights. Here, $H = \mathbb{E}_x[XX^T]$ is the input covariance and $G = \mathbb{E}_x[(\tilde{X} - X)X^T]$ is a cross-correlation matrix capturing the interaction between activation and weight errors.

Inspired by this finding, we develop a sequential column-wise quantization algorithm. For each column $k$, the quantized weight vector $\hat{W}_k$ is obtained by quantizing a corrected target that integrates our novel term with a standard error feedback mechanism:

$$\hat{W}_k = Q \left( W_k + (W_{1:(k-1)} - \hat{W}_{1:(k-1)})a_k + (WGH^{-1})_k \right) \tag{6}$$

where $Q(\cdot)$ is the quantization operator and $a_k$ are feedback coefficients. The derivation details can be found in Appendix A.3. The crucial Cumulative Error Correction term directly injects the corrective bias into the quantization process. This forces the weight quantizer $Q(\cdot)$ to be explicitly aware of the downstream activation error, steering the solution towards a global minimum of the joint error landscape. Consequently, the quantized weights are not only locally accurate but also robust to activation perturbations, preserving the feature distributions as shown in Figure 3 (bottom).

### 3.3 ADAPTIVE CORRECTION VIA BIAS-VARIANCE REGULARIZATION

While the activation-error correction term $WGH^{-1}$ is theoretically optimal, its practical application presents a critical bias-variance trade-off. As illustrated in Figure 4, a fixed, global $\alpha$ is suboptimal. Any choice $\alpha < 1$ introduces a systematic bias by under-compensating for the activation error. Conversely, an unregularized correction ($\alpha = 1$) overfits to estimation noise in the statistics $G$ and $H$ derived from finite calibration data, leading to high variance and unstable weights. We therefore

propose an adaptive method to determine an optimal regularization strength $\hat{\alpha}$ that minimizes the weight reconstruction error.

Our goal is to learn an optimal correction factor $\alpha_k$ that minimizes the layer's reconstruction loss. As detailed in Appendix A.4, by modeling the quantizer with a linear approximation, the expected loss can be decomposed into a sum of per-column objectives. Optimizing for $\alpha_k$ becomes equivalent to minimizing the following for each column:

$$\alpha_k^\star = \arg\min_{\alpha \in \mathbb{R}} \sum_k \|(\alpha - 1)\, v_k + r_{0,k} + b_k\|_H^2 \quad (7)$$

where, $v_k \triangleq (WGH^{-1})_k$ is the ideal correction, $r_{0,k}$ is the propagated error from previous columns, $b_k$ is the quantization bias.

While this objective yields a closed-form solution $\alpha_k^*$, its sensitivity to estimation noise necessitates regularization. We therefore introduce a shrinkage:

$$\hat{\alpha}_k = (1 - \lambda_k)\alpha_k^\star, \qquad \lambda_k \in [0, 1], \quad (8)$$

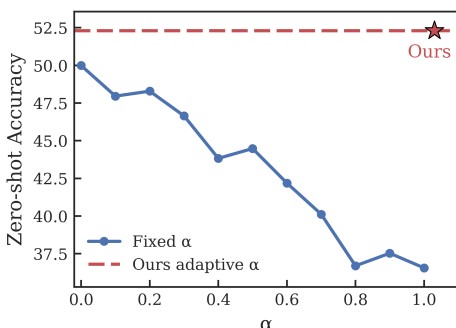

Figure 4: Model accuracy exhibits high sensitivity to the selection of a fixed, global correction factor $\alpha$. A suboptimal choice leads to significant performance degradation. Our adaptive method automatically determines a near-optimal shrinkage factor, thereby consistently outperforming any fixed $\alpha$ setting.

where $\lambda_k$ is a data-driven shrinkage factor that enhances robustness. This is implemented by introducing a data-driven ridge term $\gamma_k$ into the denominator of the solution. The final, regularized correction factor is:

$$\hat{\alpha}_k = \frac{v_k^\top (v_k - r_{0,k})}{v_k^\top v_k + \gamma_k}. \quad (9)$$

The shrinkage intensity $\lambda_k = \gamma_k/(v_k^\top v_k + \gamma_k)$ is determined automatically by the regularizer $\gamma_k$, which adaptively estimates the variance from propagated errors and baseline estimation noise. (see Appendix A.4). This adaptive regularization balances correction strength against estimation noise, ensuring stable and generalizable performance.

## 4 EXPERIMENTS

**Models and Datasets.** We apply our method to the LLaMA-2 (Touvron et al., 2023b), LLaMA-3 family and Qwen3 (Yang et al., 2025) family(8b, 14b). Following previous work, we report WikiText2 (Merity et al., 2016) perplexity (PPL) on language modeling tasks. We also perform the common sense QA evaluation on up to eight zero-shot tasks using the lm-evaluation-harness (Gao et al., 2024), including BoolQ (Clark et al., 2019), HellaSwag (Zellers et al., 2019), LAMBADA, OpenBookQA (OBQA) (Mihaylov et al., 2018), PIQA (Bisk et al., 2020), SIQA (Sap et al., 2019), WinoGrande (Sakaguchi et al., 2021), ARC-Easy, and ARC-Challenge (Boratko et al., 2018).

**Baselines and Implementation Details.** We benchmark our approach, AEC-SVQ, against SmoothQuant (Xiao et al., 2022), GPTQ (Frantar et al., 2022), OmniQuant (Shao et al., 2023), Quarot (Ashkboos et al., 2024), SpinQuant (Liu et al., 2024b), OSTQuant (Hu et al., 2025) and GPTAQ (Li et al., 2025). In AEC-SVQ, all activations are quantized using per-token asymmetric scalar quantization, while weights are quantized using vector quantization configured with 4096 centroids and a vector length of 6. We leverage the optimization methodology presented in OSTQuant to obtain the transformation **T** by minimizing the end-to-end distributional error of the hybrid SQ-VQ framework. Following established practices in weight-only vector quantization, we further fine-tune the normalization operator and the VQ codebook to enhance quantization performance. As these fine-tuned parameters constitute only a small fraction of the total layer parameters, this process is both rapid and memory-efficient.

### 4.1 OVERALL RESULTS

**Quantization Performance.** As shown in Table 1, AEC-SVQ consistently and substantially outperforms all previous state-of-the-art approaches across a diverse range of models and scales. The performance gains are particularly evident on large-scale models. For instance, On LLaMA-3 70B,

AEC-SVQ achieves a perplexity of 6.33—orders of magnitude lower than competitors like Quarot (5e4)—while simultaneously recovering over 89% of the full-precision zero-shot accuracy, showcasing a significant advance in preserving language modeling capabilities. This superiority holds across diverse architectures. On Qwen3 14B, AEC-SVQ effectively halves the perplexity error of previous SOTA while maintaining a clear lead in task accuracy.

Unlike prior methods that often trade language modeling fidelity for task performance, our approach excels at both, drastically narrowing the gap to the full-precision baseline on all fronts. Crucially, AEC-SVQ's design proves superior not only to classic methods like SmoothQuant and GPTQ but also to recent, highly sophisticated approaches such as OSTQuant and SpinQuant. These findings confirm that our hybrid scalar-vector quantization framework generalizes robustly, underscoring its broad effectiveness and applicability. More detailed results can be seen in Appendix A.5

Table 1: **Comparison of perplexity on WikiText2 and averaged accuracy on eight Zero-Shot tasks under W2A4 quantization setting.** The table shows our proposed AEC-SVQ against prominent baselines. AEC-SVQ significantly outperforms all prior methods across all models.

| | LLaMA-3 8B | | LLaMA-3 70B | | LLaMA-2 7B | | LLaMA-2 13B | | LLaMA-2 70B | | Qwen3 7B | | Qwen3 14B | |
|---|---|---|---|---|---|---|---|---|---|---|---|---|---|---|
| **Method** | 0-shot[8] | Wiki | 0-shot[8] | Wiki | 0-shot[8] | Wiki | 0-shot[8] | Wiki | 0-shot[8] | Wiki | 0-shot[8] | Wiki | 0-shot[8] | Wiki |
| (W2A4) | Avg.($\uparrow$) | ($\downarrow$) | Avg.($\uparrow$) | ($\downarrow$) | Avg.($\uparrow$) | ($\downarrow$) | Avg.($\uparrow$) | ($\downarrow$) | Avg.($\uparrow$) | ($\downarrow$) | Avg.($\uparrow$) | ($\downarrow$) | Avg.($\uparrow$) | ($\downarrow$) |
| Full Precision | 67.13 | 6.14 | 70.59 | 3.32 | 64.15 | 5.47 | 66.48 | 4.88 | 70.59 | 3.32 | 67.28 | 9.72 | 70.32 | 8.65 |
| SmoothQuant | 35.22 | 1e6 | 34.28 | 7e5 | 34.41 | 5e5 | 34.88 | 2e5 | 34.18 | 2e5 | 34.98 | 3e5 | 35.48 | 6e5 |
| OmniQuant | 36.46 | 2e6 | 34.11 | 6e5 | 33.69 | 4e5 | 35.36 | 1e5 | 34.89 | 9e4 | 34.63 | 3e5 | 36.23 | 5e5 |
| QuaRot | 36.31 | 3e5 | 35.42 | 5e4 | 36.06 | 1e5 | 36.28 | 8e5 | 34.17 | 8e3 | 35.52 | 1e5 | 37.04 | 3e5 |
| SpinQuant+GPTQ | 36.69 | 96.94 | 35.72 | 3e5 | 38.45 | 124.79 | 41.85 | 23.64 | 45.91 | 656.00 | 44.16 | 24.57 | 41.61 | 41.57 |
| SpinQuant+GPTAQ | 40.37 | 48.31 | 36.80 | 4e5 | 38.69 | 7e3 | 42.28 | 33.21 | 52.57 | 200.00 | 43.20 | 25.24 | 47.49 | 17.04 |
| OSTQuant+GPTQ | 38.33 | 36.20 | 38.33 | 618.90 | 36.35 | 41.15 | 43.95 | 15.85 | 49.99 | 11.31 | 42.82 | 27.49 | 52.90 | 17.55 |
| OSTQuant+GPTAQ | 41.05 | 20.20 | 38.29 | 559.68 | 42.47 | 12.46 | 46.84 | 8.90 | 57.17 | 7.71 | 46.12 | 24.62 | 51.77 | 17.51 |
| **AEC-SVQ** | **58.39** | **8.65** | **63.11** | **6.33** | **56.20** | **6.29** | **60.78** | **5.49** | **65.63** | **4.41** | **60.52** | **11.27** | **63.70** | **10.38** |

**Speedup and memory savings.** Our AEC-SVQ framework yields substantial improvements in inference efficiency, as detailed in Table 2. The method dramatically reduces the memory footprint, with savings factors peaking at over 7.0x for common short sequence lengths. While this advantage naturally moderates with longer contexts, the memory reduction remains highly effective, exceeding 3.5x across all models even at a sequence length of 8192, underscoring its value in memory-constrained scenarios.

In addition to memory optimization, AEC-SVQ provides robust prefill acceleration. The speedup consistently surpasses 2.2x across most configurations and scales positively with model size, reaching up to 3.612x for the 30B model. This sustained acceleration, combined with the significant memory savings, confirms that our method makes the deployment of large models more computationally practical and efficient without compromising performance.

Table 2: **Prefill speedup and memory saving factor of AEC-SVQ.** Measurements are conducted on LLaMA models with different parameter sizes and sequence lengths. All tests were conducted on a Transformer block with batch size 4 on a 3090 GPU. Refer to Appendix A.5.2 for more details.

| Model Size | Prefill Speedup (Seqlen) | | | | | | Memory Saving Factor (Seqlen) | | | | | |
|---|---|---|---|---|---|---|---|---|---|---|---|---|
| | 256 | 512 | 1024 | 2048 | 4096 | 8192 | 256 | 512 | 1024 | 2048 | 4096 | 8192 |
| 7B | 2.420× | 2.334× | 2.235× | 2.224× | 2.207× | 1.730× | 6.266× | 5.902× | 5.367× | 4.728× | 4.103× | 3.615× |
| 8B | 2.666× | 2.575× | 2.621× | 2.462× | 2.375× | 2.263× | 6.317× | 6.051× | 5.613× | 4.991× | 4.273× | 3.593× |
| 13B | 2.806× | 2.909× | 2.764× | 2.566× | 2.848× | 2.333× | 6.686× | 6.326× | 5.730× | 5.096× | 4.372× | 3.799× |
| 30B | 3.612× | 3.177× | 3.054× | 3.450× | 2.860× | 2.682× | 7.082× | 6.699× | 6.197× | 5.493× | 4.697× | 4.029× |

## 4.2 ABLATION STUDY

**Ablation on AEC-SVQ.** We conduct a comprehensive ablation study to validate the effectiveness of each component in our proposed AEC-SVQ framework, as shown in Table 3. The study confirms the superiority of an optimized transformation matrix, as our proposed learned transformation improves perplexity to 12.13 and boosts accuracy to 48.24. Building upon this, the introduction of CEAVQ and the adaptive correction factor $\alpha$ provides further incremental refinements to both

Table 3: **Ablation study on the components of AEC-SVQ.** Starting from a baseline weight-activation scalar quantization (W-A-SQ), we progressively integrate our key contributions. Avg Acc denotes the average accuracy over zero-shot[5] All results are reported on the LLaMA-3 8B model.

| Method | Wiki($\downarrow$) | Avg Acc ($\uparrow$) |
|---|---|---|
| W-A-SQ | *NaN* | 34.49 |
| + Weight VQ | 2348.20 | 35.83 |
| + Codebook Quantization | 2519.75 | 35.50 |
| + Local Reconstruction | 1405.55 | 35.70 |
| + Hadamard Transformation | 14.97 | 46.79 |
| + Learned Transformation | 12.13 | 48.24 |
| + CEAVQ ($\alpha$=0.25) | 11.42 | 49.94 |
| + Adaptive $\alpha$ | 10.46 | 52.31 |
| **+ Fine-tune** | **8.65** | **58.39** |

Table 4: **Performance of AEC-SVQ on 2-bit weight-only quantization.** To highlight the general applicability of our framework, we adapt it to a weight-only quantization setting. AEC-SVQ outperforms methods designed specifically for this task, demonstrating its superior performance.

| Method | Wiki($\downarrow$) | Avg Acc($\uparrow$) |
|---|---|---|
| Full Precision | 6.14 | 72.81 |
| GPTQ | 210.00 | 36.16 |
| DB-LLM | 13.60 | 51.74 |
| QuIP | 85.10 | 36.81 |
| QuIP# | 9.11 | - |
| VPTQ | 9.29 | 60.22 |
| PCDVQ | 8.77 | 58.60 |
| **AEC-SVQ (ours)** | **8.02** | **64.01** |

metrics. Subsequently, the final fine-tuning step delivers another substantial performance leap. This step-by-step analysis demonstrates that each component of AEC-SVQ plays a crucial and cumulative role in achieving its final state-of-the-art performance.

**AEC-SVQ for weight only quantization.** To demonstrate its versatility, we adapt the AEC-SVQ framework to the 2-bit weight-only quantization setting. As shown in Table 4, our method outperforms specialized state-of-the-art techniques in this domain. AEC-SVQ achieves a leading average accuracy of 64.01, while attaining the lowest perplexity of 8.02. This strong performance in a distinct quantization paradigm, achieved without fundamental modifications, underscores the robustness and generality of our core framework for minimizing quantization error.

Table 5: **Ablation study on the fine-tuning process for our W2A4 model.** Starting from a no-FT baseline, we systematically explore the impact of tuning different parameters, using different optimizers, and applying various learning rate schedules.

| Method | Bits | FT params | FT LR | Dataset | Wiki-PPL $\downarrow$ | Zero-shot[8] $\uparrow$ |
|---|---|---|---|---|---|---|
| FP32 (full precision) | FP32 | – | – | – | 6.14 | 67.13 |
| no FT | W2A4 | – | – | – | 10.46 | 52.31 |
| + FT (Adam) | W2A4 | layernorm | 5e−5 | Wiki+C4 | 9.62 | 51.50 |
| + FT (Adam) | W2A4 | layernorm | 5e−5 | RedPajama | 10.05 | 52.36 |
| + FT (AdamW) | W2A4 | layernorm | 5e−5 | Wiki+C4 | 9.34 | 54.58 |
| + FT (AdamW) | W2A4 | layernorm + VQ codebook | LN=5e−5; CB=5e−5 | Wiki+C4 | 10.19 | 53.95 |
| + FT (AdamW) | W2A4 | layernorm + VQ codebook | LN=5e−5; CB=1e−5 | Wiki+C4 | 8.81 | 58.18 |
| + FT (AdamW) | W2A4 | layernorm + VQ codebook | LN=1e−5; CB=5e−6 | Wiki+C4 | **8.65** | **58.39** |

**Ablation on Fine-tuning.** We perform a detailed ablation on the post-quantization fine-tuning (FT) process to identify the optimal strategy, with results in Table 5. Our analysis reveals that while tuning only the LayerNorm offers moderate gains, co-tuning the VQ codebook is critical, providing a substantial boost to zero-shot accuracy. Furthermore, we found that applying distinct learning rates—specifically was superior to a uniform schedule.

## 5 CONCLUSION

In this paper, we introduce AEC-SVQ, a novel hybrid framework designed to resolve the fundamental trade-off between computational efficiency and memory compression for ultra-low-bit W2A4 LLM inference. Our approach is built on three synergistic innovations. First, we propose a **hybrid SQ-VQ scheme** centered on a single learned transformation that simultaneously optimizes data distributions for weight VQ, activation SQ, and codebook quantization. Second, our **Cumulative-Error-Aware VQ (CEAVQ)** algorithm introduces a principled method to proactively compensate for compounding errors by aligning the quantized layer's output with its full-precision distribution. Finally, we develop an **Adaptive Compensation** mechanism that uses a closed-form, data-driven shrinkage factor to ensure robustness against statistical noise from limited calibration data. Extensive experiments demonstrate that AEC-SVQ consistently outperforms existing state-of-the-art quantization methods. These results validate the effectiveness of our integrated approach and establish a new frontier for LLM quantization, making high-performance, ultra-low-bit models practical for deployment in resource-constrained environments.

## 6  ETHICS STATEMENT

This work does not involve human subjects, personal data, or sensitive content. It focuses solely on optimization of LLM compression and inference. Therefore, we believe it does not raise ethical concerns.

## 7  REPRODUCIBILITY STATEMENT

We provide complete details of our algorithms and evaluation protocols in the main paper and appendix. All models are evaluated on publicly available benchmarks (Wikitext-2, ARC, BoolQ, PIQA, HellaSwag, OBQA, SIQA, and WinoGrande). The code for our algorithms and reproducing experiments will be released upon publication. These resources will ensure full reproducibility of the reported results.

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

## A APPENDIX

### A.1 LLM DISCLAIMER

The authors hereby declare the role of large language model (LLM) tools in the preparation of this manuscript: LLMs were solely utilized to assist with text polishing (including refining sentence structure, optimizing lexical expression, and enhancing language fluency) and **writing optimization** of the paper's narrative content.

It is explicitly emphasized that all core components of this research, which determine the originality, scientific validity, and academic value of the work, were independently completed by the research team through manual efforts. These components include, but are not limited to:

- The formulation and development of the overall research framework, core ideas, and logical structure of the study;
- The design, coding, debugging, and validation of all algorithms and program codes involved in the research;
- The design of experimental protocols, collection and preprocessing of experimental data, execution of experiments, analysis and interpretation of experimental results, and verification of conclusions.

The use of LLM tools did not involve any participation in the conception of research content, generation of technical solutions, implementation of experimental processes, or derivation of research conclusions. All content of this paper adheres to academic integrity standards, and the research team assumes full responsibility for the scientificity, authenticity, and originality of the work.

## A.2 THEORETICAL ANALYSIS OF HYBRID SQ-VQ FRAMEWORK WITH LEARNED TRANSFORMATION

### A.2.1 PRELIMINARIES AND NOTATION

Consider a single linear layer

$$y = Wx \in \mathbb{R}^{d_{\text{out}}}, \qquad x \in \mathbb{R}^d (d = d_{\text{in}}),$$

with input second moment (covariance) $H = \mathbb{E}[xx^\top] \succ 0$. We analyze three sources of quantization error applied around this layer: (i) *activation scalar quantization* (SQ) of $x$, (ii) *weight vector quantization* (VQ) of $W$ after reshaping into length-$p$ vectors (with $p = \text{vec\_len}$), and (iii) *codebook integer quantization* of the learned VQ codewords.

We insert a *function-preserving equivalent transformation pair*

$$T = \Lambda O, \qquad \Lambda = \text{diag}(\lambda_1, \ldots, \lambda_d) \succ 0, O \in \mathbb{O}(d),$$

in the sense that we work in the primed coordinates

$$x' = Tx, \qquad W' = WT^{-1},$$

so that $Wx = W'x'$ in floating point. Quantization is performed in the primed coordinates and the pair can be fused away at deployment. Throughout, we adopt the standard high-resolution/small-noise approximation and assume independence between signals and quantization noises when taking second-order moments.

**Weight reshaping for VQ.** Following the setting in the main paper, we reshape the weight matrix $W' \in \mathbb{R}^{d_{\text{out}} \times d}$ along the input dimension into an i.i.d.-like collection of vectors $\{z_i' \in \mathbb{R}^p\}_{i=1}^N$ (row blocks or columnwise chunks of length $p = \text{vec\_len}$). K-means with $K = \text{num\_centroids}$ produces a codebook $\mathcal{C} = \{c_k'\}_{k=1}^K \subset \mathbb{R}^p$ and assignments $\pi(i) \in [K]$.

### A.2.2 UNIFIED SECOND-ORDER ERROR MODEL

Let $\eta_x$ denote the SQ error on activations, $\eta_w$ the VQ reconstruction error on weights, and $\eta_c$ the additional error stemming from integer quantization of codewords (propagated back to the weight domain). In primed coordinates we write $\eta_x', \eta_w', \eta_c'$ with covariances

$$\Sigma_{\eta_x'} = \mathbb{E}[\eta_x' \eta_x'^\top], \Sigma_{\eta_w'} = \mathbb{E}[\eta_w' \eta_w'^\top], \Sigma_{\eta_c'} = \mathbb{E}[\eta_c' \eta_c'^\top].$$

Neglecting second-order cross terms (small-noise linearization), the output error obeys

$$\hat{y} - y \approx W'\eta_x' + (\eta_w' + \eta_c') \, x'.$$

Taking squared norm and expectation, using independence between $x'$ and weight-side noises, we obtain the *unified trace form*

$$\mathcal{E} = \mathbb{E}\|\hat{y} - y\|_2^2 \approx \underbrace{\text{tr}\big(W'^\top W' \Sigma_{\eta_x'}\big)}_{\text{(A) activation-side propagation}} + \underbrace{\text{tr}\big(H' \left(\Sigma_{\eta_w'} + \Sigma_{\eta_c'}\right)\big)}_{\text{(B) weight-side propagation}}, \tag{10}$$

where $H' = \mathbb{E}[x'x'^\top] = THT^\top$. Thus, any transformation $T$ that jointly reduces the spectra/diagonals of $\Sigma_{\eta_x'}$, $\Sigma_{\eta_w'}$, and $\Sigma_{\eta_c'}$ tends to decrease the unified error $\mathcal{E}$.

### A.2.3 NOISE MODELS IN THE PRIMED COORDINATES

**(i) Activation SQ.** For uniform mid-rise/tread scalar quantizers with per-axis step sizes $\Delta_j$ and negligible overload,

$$\Sigma_{\eta_x'} \approx \text{diag}\left(\frac{\Delta_1^2}{12}, \ldots, \frac{\Delta_d^2}{12}\right), \qquad \Delta_j \propto \alpha_j,$$

where $\alpha_j$ is the (symmetric) dynamic range bound on the $j$-th coordinate of $x'$. Large coordinates (heavy tails, outliers) directly inflate $\Delta_j$.

**(ii) Weight VQ (k-means).** Let $z_i' \in \mathbb{R}^p$ be the reshaped weight vectors extracted from $W'$. K-means yields reconstructions $\hat{z}_i' = c'_{\pi(i)}$ and errors $e_i' = z_i' - \hat{z}_i'$. Under high-rate assumptions and for subGaussian/Gaussian-like vector statistics, the mean distortion per vector obeys the Zador/Gersho-type scaling

$$\mathbb{E}\|e'\|_2^2 \approx C_p|\Sigma_{z'}|^{1/p}K^{-2/p}, \tag{11}$$

where $\Sigma_{z'}$ is the empirical covariance of the block distribution $\{z_i'\}$, and $C_p$ depends only on the dimension $p$ and optimal cell shape.

**(iii) Codebook integer quantization.** After VQ, each codeword $c_k' \in \mathbb{R}^p$ is uniformly quantized per coordinate to a $b$-bit integer grid with shared (or per-dimension shared) step size $\Delta_c$. For negligible overload,

$$\Sigma_{\eta_c'} \approx \frac{\Delta_c^2}{12} I_p(\text{per block}), \qquad \Delta_c \propto \alpha_c, \alpha_c = \max_{k,j}|(c_k')_j|.$$

Thus $\alpha_c$—the $\ell_\infty$ radius of the codebook cloud along coordinate axes—controls the integer quantization error.

### A.2.4 Two Elementary Lemmas on Whitening and Energy Equalization

**Lemma 1** (Whitening optimality for second-order criteria). *Let $x$ be subGaussian with covariance $H \succ 0$. Consider $T_{\text{wh}} = \Lambda_{\text{wh}}O$ with $\Lambda_{\text{wh}} = H^{-1/2}$ and any $O \in \mathbb{O}(d)$. Then $x' = T_{\text{wh}}x$ satisfies $\text{Cov}(x') = I$. Among all linear transforms with fixed trace of the output covariance, whitening equalizes all eigenvalues, and hence minimizes the geometric mean of eigenvalues:*

$$\prod_{j=1}^{p} \lambda_j(\Sigma_{z'}) \text{ is minimized when } \Sigma_{z'} \propto I_p.$$

*Consequently, for block statistics derived from right-multiplying $W$ by $T_{\text{wh}}^{-1}$, the high-rate VQ proxy $|\Sigma_{z'}|^{1/p}$ is minimized.*

*Proof.* By construction, $\Lambda_{\text{wh}} = H^{-1/2}$ equalizes the eigenvalues of the output covariance to 1. For any positive semidefinite matrix with fixed trace, AM$\geq$GM implies that the geometric mean of eigenvalues is minimized when all eigenvalues are equal. The claimed consequence for the proxy $|\Sigma_{z'}|^{1/p}$ follows from the monotonicity of equation 11 in $|\Sigma_{z'}|^{1/p}$. □

**Lemma 2** (Energy-equalizing rotations minimize the $\ell_\infty$ magnitude). *For any nonzero $u \in \mathbb{R}^d$,*

$$\min_{O \in \mathbb{O}(d)} \|Ou\|_\infty = \frac{\|u\|_2}{\sqrt{d}}.$$

*In particular, there exists $O^\star$ such that $O^\star u = \|u\|_2\, d^{-1/2}\, s$ for some $s \in \{\pm 1\}^d$, i.e., all coordinates have equal magnitude. Moreover, for subGaussian $x'$, one obtains*

$$\mathbb{E}\|O^\star x'\|_\infty \leq \frac{1}{\sqrt{d}}\,\mathbb{E}\|x'\|_2 \text{ and } \|O^\star x'\|_\infty \leq \frac{\|x'\|_2}{\sqrt{d}} \text{ samplewise}.$$

*Proof.* For any $O$, $\|Ou\|_\infty \geq \|Ou\|_2/\sqrt{d} = \|u\|_2/\sqrt{d}$ by norm inequalities; hence $\inf_O \|Ou\|_\infty \geq \|u\|_2/\sqrt{d}$. Equality is achieved by taking any orthogonal $O^\star$ that maps the unit vector $u/\|u\|_2$ to the constant-sign vector $d^{-1/2}s$ (both are unit-norm), which exists because the orthogonal group acts transitively on the unit sphere. The sub-Gaussian bound follows immediately. □

### A.2.5 Main Proposition: A Single $T = \Lambda O$ Benefits All Three Quantizers

**proposition 1** (Joint improvement under a learnable rotation–smooth transform). *Assume the high-resolution regime with negligible overload and independence between signals and quantization noises. Let*

$$T^\star = \underbrace{H^{-1/2}}_{\Lambda^\star} \underbrace{O^\star}_{\text{energy equalization}},$$

*where $O^\star$ is any orthogonal transform that approximately equalizes coordinate magnitudes (e.g., a Hadamard-like rotation or a learned orthogonal matrix). Then the unified error $\mathcal{E}$ in equation 10 strictly decreases compared to $T = I$:*

$$\Delta\mathcal{E} = \mathcal{E}(T^\star) - \mathcal{E}(I) < 0.$$

*In particular, each constituent term decreases:*

*Activation SQ:*      $\Sigma_{\eta'_x} = \mathrm{diag}(\Delta_1^2/12, \dots), \Delta_j \propto \alpha_j(T^\star) \downarrow \Rightarrow \mathrm{tr}(W'^\top W' \Sigma_{\eta'_x}) \downarrow,$

*Weight VQ:*      $\left|\Sigma_{z'}\right|^{1/p} \downarrow \Rightarrow \mathbb{E}\|e'\|_2^2 \approx C_p\left|\Sigma_{z'}\right|^{1/p} K^{-2/p} \downarrow,$

*Codebook int-quant:*      $\alpha_c(T^\star) = \max_{k,j} |(c'_k)_j| \downarrow \Rightarrow \Sigma_{\eta'_c} \propto \Delta_c^2 \downarrow.$

*Proof sketch. Step 1 (VQ via whitening).* By Lemma 1, $\Lambda^\star = H^{-1/2}$ equalizes second-order statistics in the primed coordinates, driving block covariances towards $\Sigma_{z'} \propto I_p$ and thereby minimizing the high-rate proxy $|\Sigma_{z'}|^{1/p}$. Hence the mean VQ distortion decreases.

*Step 2 (SQ and codebook via $\ell_\infty$ control).* By Lemma 2, for each sample of $x'$ the energy-equalizing rotation $O^\star$ enforces $\|O^\star x'\|_\infty \le \|x'\|_2/\sqrt{d}$. Therefore the per-axis dynamic ranges $\alpha_j(T^\star)$ contract by a factor on the order of $1/\sqrt{d}$, enabling uniformly smaller steps $\Delta_j$ for SQ and reducing $\Sigma_{\eta'_x}$. The same $\ell_\infty$ contraction applies to codeword coordinates (by the same energy-equalization principle acting on block vectors), shrinking the global codebook bound $\alpha_c(T^\star)$ and thus $\Delta_c$.

*Step 3 (Monotonicity in the unified trace).* Each of the three covariance terms decreases in the Loewner order (or at least in trace), so both traces in equation 10 decrease, which implies $\Delta\mathcal{E} < 0$.    □

### A.2.6   REMARKS ON "SMOOTH" (MIXING) TRANSFORMS AND TAILS

Beyond orthogonal rotations, one may allow light smoothing/mixing (still linear and invertible) inside $T$ to average multiple coordinates per output coordinate. Under standard subGaussian/CLT heuristics, this further reduces kurtosis and extreme-value behavior, lowering overload probabilities for SQ and tightening the extreme codeword coordinate $\alpha_c$. Such smoothing can be learned jointly with $O$ while maintaining the factorization $T = \Lambda O$ (with $\Lambda$ diagonal and $O$ orthogonal) by absorbing any additional conditioning into $\Lambda$ and keeping the remainder orthogonal.

### A.2.7   ASSUMPTIONS AND LIMITATIONS

The analysis rests on (i) high-resolution quantization (overload negligible after appropriate clipping), (ii) small-noise linearization (neglecting cross terms), and (iii) subGaussian or light-tailed statistics enabling the proxies equation 11. In practice, learnable $T$ can be trained end-to-end to approximate $H^{-1/2}$ and energy-equalizing rotations; the proposition guarantees the existence of such a beneficial transform and explains its joint effect on the three quantizers.

## A.3   DERIVATION OF CEAVQ

### A.3.1   WEIGHT-ACTIVATION QUANTIZATION PROXY OBJECTIVE

We study the proxy loss

$$\ell(\hat{W}) = \mathbb{E}_X\left[\left\|\hat{W}X - W\tilde{X}\right\|_F^2\right] = \mathbb{E}_X\left[\left\|(\hat{W} - W)X - W(\tilde{X} - X)\right\|_F^2\right], \tag{12}$$

where $W, \hat{W} \in \mathbb{R}^{m \times n}$ are the full-precision and quantized weight matrices, respectively, and $\tilde{X}, X \in \mathbb{R}^{n \times p}$ denote a floating-point input and its (possibly stochastic) quantized counterpart. The expectation is taken with respect to the randomness of $X$ (and hence $\tilde{X}$ when it is a function of $X$). Throughout we use $\|A\|_F^2 = \mathrm{tr}(A^\top A)$ and the cyclic property of the trace, $\mathrm{tr}(ABC) = \mathrm{tr}(BCA)$, whenever dimensions are compatible. We assume $\mathbb{E}\|X\|_F^2 < \infty$ so that all traces are finite.

**Step 1: Quadratic expansion.** Define

$$A \triangleq (\hat{W} - W)X - W(\tilde{X} - X), \qquad B \triangleq (\hat{W} - W)X, \qquad C \triangleq W(\tilde{X} - X),$$

so $A = B - C$. Then

$$\|A\|_F^2 = \mathrm{tr}\big[(B - C)^\top(B - C)\big] = \mathrm{tr}(B^\top B) - 2\,\mathrm{tr}(B^\top C) + \mathrm{tr}(C^\top C). \tag{13}$$

Taking expectations and using linearity of $\mathbb{E}$ yields

$$\ell(\hat{W}) = \mathbb{E}\big[\mathrm{tr}(B^\top B)\big] - 2\,\mathbb{E}\big[\mathrm{tr}(B^\top C)\big] + \mathbb{E}\big[\mathrm{tr}(C^\top C)\big]. \tag{14}$$

**Step 2: Move fixed matrices outside the expectation.** Because $W$ and $\hat{W}$ are deterministic (conditioned on the current layer),

$$\mathbb{E}\big[\mathrm{tr}(B^\top B)\big] = \mathbb{E}\Big[\mathrm{tr}\Big(X^\top(\hat{W} - W)^\top(\hat{W} - W)X\Big)\Big] = \mathrm{tr}\Big((\hat{W} - W)^\top(\hat{W} - W)\,\mathbb{E}[XX^\top]\Big), \tag{15}$$

$$\mathbb{E}\big[\mathrm{tr}(B^\top C)\big] = \mathbb{E}\Big[\mathrm{tr}\Big(X^\top(\hat{W} - W)^\top W(\tilde{X} - X)\Big)\Big] = \mathrm{tr}\Big((\hat{W} - W)^\top W\,\mathbb{E}[(\tilde{X} - X)X^\top]\Big), \tag{16}$$

$$\mathbb{E}\big[\mathrm{tr}(C^\top C)\big] = \mathbb{E}\Big[\mathrm{tr}\Big((\tilde{X} - X)^\top W^\top W(\tilde{X} - X)\Big)\Big] = \mathrm{tr}\Big(W^\top W\,\mathbb{E}[(\tilde{X} - X)(\tilde{X} - X)^\top]\Big). \tag{17}$$

**Step 3: Collect second-order statistics.** Introduce the second-order moment matrices

$$H \triangleq \mathbb{E}[XX^\top], \qquad G \triangleq \mathbb{E}[(\tilde{X} - X)X^\top], \qquad K \triangleq \mathbb{E}[(\tilde{X} - X)(\tilde{X} - X)^\top]. \tag{18}$$

Substituting equation 15–equation 17 into equation 14 gives the compact form

$$\ell(\hat{W}) = \mathrm{tr}\Big((\hat{W} - W)^\top(\hat{W} - W)\,H\Big) - 2\,\mathrm{tr}\Big((\hat{W} - W)^\top W\,G\Big) + \mathrm{tr}\Big(W^\top W\,K\Big). \tag{19}$$

**Remarks.** (i) The third term in equation 19 is independent of $\hat{W}$ and hence acts as a constant offset when optimizing over $\hat{W}$ (given fixed $W$ and an input quantizer determining $K$). (ii) The first term weighs the weight-quantization error $(\hat{W} - W)$ by the input second moment $H$, while the middle term captures the interaction between weight and activation quantization through $G$.

### A.3.2 COMPLETING THE SQUARE AND THE UNCONSTRAINED MINIMIZER

Let $\Delta W \triangleq \hat{W} - W$ and define the $H$-weighted inner product $\langle A, B\rangle_H \triangleq \mathrm{tr}(A^\top B H)$. Then equation 12 reads

$$\ell(\hat{W}) = \langle \Delta W, \Delta W\rangle_H - 2\langle \Delta W, WGH^{-1}\rangle_H + \mathrm{tr}(W^\top W K).$$

Completing the square under $\langle \cdot, \cdot\rangle_H$ yields

$$\ell(\hat{W}) = \big\langle \Delta W - WGH^{-1}, \Delta W - WGH^{-1}\big\rangle_H + C, \tag{20}$$

with the constant

$$C = \mathrm{tr}(W^\top W K) - \big\langle WGH^{-1}, WGH^{-1}\big\rangle_H = \mathrm{tr}(W^\top W K) - \mathrm{tr}\big((WGH^{-1})^\top(WGH^{-1})H\big). \tag{21}$$

The (unconstrained) minimizer is therefore

$$\hat{W}^\star = W + WGH^{-1}, \tag{22}$$

which coincides with the stationarity condition $\nabla_{\hat{W}}\ell(\hat{W}) = 2\big((\hat{W} - W)H - WG\big) = 0$. In practice, $\hat{W}$ must lie in a quantized space; we will approach equation 22 iteratively.

### A.3.3    LDL$^{\top}$-STYLE COLUMNWISE DECOMPOSITION

To expose a columnwise structure, factor the (symmetric) matrix $H$ as

$$H = (U + I)\, D\, (U + I)^{\top}, \tag{23}$$

where $U$ is strictly upper triangular and $D = \mathrm{diag}(d_1, \ldots, d_n) \succcurlyeq 0$. (Equivalently, $H = LDL^{\top}$ with $L = (U + I)^{\top}$ unit lower triangular.) For any $m \times n$ matrix $M$ denote its $k$th column by $M_k$ and the strict prefix by $M_{1:(k-1)}$. Using equation 23 and the cyclic property of the trace,

$$\langle \Delta W, \Delta W \rangle_H = \mathrm{tr}\big((\Delta W(U+I))^{\top}\, \Delta W(U+I)\, D\big) = \sum_{k=1}^{n} d_k \left\| \Delta W(U+I)e_k \right\|_F^2$$

$$= \sum_{k=1}^{n} d_k \left\| \Delta W_k + \Delta W_{1:(k-1)}\, u_k \right\|_F^2, \tag{24}$$

where $u_k \triangleq U_{1:(k-1),k} \in \mathbb{R}^{k-1}$ collects the $k$th column of $U$ above the diagonal. Replacing $\Delta W$ by $\Delta W - WGH^{-1}$ per equation 20 yields the column-coupled objective

$$\sum_{k=1}^{n} d_k \left\| \underbrace{(\hat{W}_k - W_k)}_{\text{current column}} + \underbrace{(\hat{W}_{1:(k-1)} - W_{1:(k-1)})}_{\text{previously fixed}} u_k - \underbrace{\left((WGH^{-1})_k + (WGH^{-1})_{1:(k-1)}\, u_k\right)}_{\text{cross-term compensation}} \right\|_F^2 + C. \tag{25}$$

Given $\hat{W}_{1:(k-1)}$, the $k$th subproblem is a (weighted) least-squares fit of $\hat{W}_k$ to an *effective target*

$$T_k^{\text{exact}} = W_k + \big(W_{1:(k-1)} - \hat{W}_{1:(k-1)}\big)u_k + (WGH^{-1})_k + (WGH^{-1})_{1:(k-1)}\, u_k. \tag{26}$$

If the quantizer $Q(\cdot)$ is fixed (e.g., a vector quantizer with a frozen codebook), the greedy update is simply $\hat{W}_k \leftarrow Q\big(T_k^{\text{exact}}\big)$.

**Practical simplification.**   To reduce overhead, we often approximate equation 26 by retaining the dominant self-compensation term $(WGH^{-1})_k$ and absorb the history-dependent $(WGH^{-1})_{1:(k-1)} u_k$ into the feedback through $u_k$ (or damp it with a scalar). This leads to

$$T_k = W_k + \big(W_{1:(k-1)} - \hat{W}_{1:(k-1)}\big)a_k + (WGH^{-1})_k, \tag{27}$$

where $a_k \in \mathbb{R}^{k-1}$ is a feedback vector (default $a_k = u_k$).

### A.3.4    COUPLED DECOMPOSITIONS FOR $H$ AND $G$

When feasible, we align $G$ with the same triangular basis induced by $H$ by seeking

$$G \approx (U + I)\, D_G\, (U + I)^{\top}, \tag{28}$$

with $D_G$ (approximately) diagonal. One practical choice is to set $U$ from the exact $LDL^{\top}$ of $H$ and define $D_G \triangleq \mathrm{diag}\big((U+I)^{-1}G(U+I)^{-\top}\big)$ (componentwise on the diagonal), discarding off-diagonal residuals.[1] This alignment causes the cross-term $WGH^{-1}$ to predominantly affect the columnwise targets via the terms already present in equation 27, improving stability of the greedy updates.

### A.3.5    GREEDY COLUMNWISE UPDATE WITH CROSS-TERM FEEDBACK

With the above ingredients, our adaptive quantization step for column $k$ is

$$\hat{W}_k = Q\Big(W_k + (W_{1:(k-1)} - \hat{W}_{1:(k-1)})\, a_k + (WGH^{-1})_k\Big). \tag{29}$$

This is akin to LDLQ-style feedback, augmented by a linear-term compensation $(WGH^{-1})_k$ that explicitly targets the shift in the completed square equation 20. Unless stated otherwise, we set $a_k = u_k$ from equation 23 and $\alpha = 0.25$. In our implementation, $Q(\cdot)$ updates only the VQ assignment indices while keeping the codebook fixed.

---

[1] Since $G$ need not be symmetric, one may use its symmetrization $\frac{1}{2}(G + G^{\top})$ for this projection.

**Choice of $a_k$ and $U$.** Given $H = (U + I)D(U + I)^\top$, $a_k = u_k$ is the optimal feedback in the sense that it exactly decouples the quadratic term equation 24 into a sum of per-column $\ell_2$ objectives. If $H$ is ill-conditioned, we compute $U, D$ via a pivoted $LDL^\top$ (or Cholesky) factorization of $H + \lambda I$ with a small $\lambda$.

### A.3.6 NOISE SHAPING VIA $K$ AND ITS ONLINE REFINEMENT

Let $\eta \triangleq Q(z) - z$ denote the quantization error applied elementwise to a vector $z$. Its covariance $K = \mathbb{E}[\eta\eta^\top]$ enters equation 12 only through the constant $C$. Nevertheless, to keep the model consistent with the evolving targets equation 29, we update an online estimate $\widehat{K}$ using mini-batch residuals $\hat{\eta}$ observed during quantization:

$$\widehat{K} \leftarrow \beta\,\widehat{K} + (1 - \beta)\,\widehat{\mathrm{Cov}}(\hat{\eta}), \qquad \beta \in [0, 1), \tag{30}$$

and optionally shape the error in the $(U + I)$-basis so that its dominant directions align with the (projected) cross-statistics in §A.3.4. This reduces the effective linear term through better agreement between $G$ and the realized noise.

### A.3.7 ALGORITHM

1. **Initialization.** Set $\hat{W} \leftarrow 0$ (or $\hat{W} \leftarrow Q(W)$). Estimate $H = \mathbb{E}[XX^\top]$ and (optionally) $G = \mathbb{E}[(\tilde{X} - X)X^\top]$ and $K$. Compute the factorization $H = (U + I)D(U + I)^\top$; set $a_k \leftarrow u_k$ for $k = 1, \ldots, n$.

2. **For $k = 1$ to $n$** (columnwise quantization):

   (a) **Feedback computation:**
   $$\Delta_k = (W_{1:(k-1)} - \hat{W}_{1:(k-1)})\,a_k + (WGH^{-1})_k.$$

   (b) **Quantize column:**
   $$\hat{W}_k \leftarrow Q(W_k + \Delta_k) \quad \text{(update VQ indices only; no codebook re-training).}$$

3. **Feedback refinement (optional).** If using the coupled projection equation 28, periodically recompute $a_k$ (through $U$) and adjust $\alpha$ to maintain descent (see below).

4. **Noise update (optional).** Update $\widehat{K}$ via equation 30.

### A.4 DETAILED DERIVATION OF THE OPTIMAL ADAPTIVE CORRECTION FACTOR WITH BIAS-VARIANCE REGULARIZATION

Recall the completed-square form of the proxy objective:

$$\ell(\hat{W}) = \mathrm{tr}\Big((\hat{W} - W - V)^\top(\hat{W} - W - V)\,H\Big) + C, \quad V \triangleq WGH^{-1}, \tag{31}$$

with $C = \mathrm{tr}(W^\top WK) - \mathrm{tr}(V^\top VH)$ independent of $\hat{W}$. We quantize columnwise using the target

$$\hat{W}_k = Q\Big(W_k + r_{0,k} + \alpha\,v_k\Big), \qquad r_{0,k} \triangleq (W_{1:(k-1)} - \hat{W}_{1:(k-1)})\,a_k, \quad v_k \triangleq (WGH^{-1})_k. \tag{32}$$

**Bias.** When $\alpha < 1$, the applied correction $\alpha\,v_k$ shrinks the ideal correction $v_k$, producing a systematic under-compensation even if $H$ and $G$ are known exactly. Hence the expected loss cannot attain the unconstrained minimum at $\hat{W}^\star = W + V$.

**Variance.** In practice, $H$ and $G$ are estimated from a finite calibration set, yielding $H_{\mathrm{est}}$ and $G_{\mathrm{est}}$ and a random correction $WG_{\mathrm{est}}H_{\mathrm{est}}^{-1}$. Its variance typically decays with sample size but can be large for small/heteroskedastic datasets, so $\alpha = 1$ may overfit calibration noise. Choosing $\alpha < 1$ acts as a shrinkage factor that reduces variance (at the cost of bias), improving generalization of the quantized weights.

**Interpretation.** Thus $\alpha$ plays the role of a regularization parameter: $\alpha = 1$ corresponds to an unregularized, MLE-like plug-in solution; $\alpha = 0$ ignores the cross-term $G$ and reverts to a purely weight-error objective. Our goal is to select (per layer, or per column) an $\alpha$ on the regularization path that maximizes downstream generalization.

### A.4.1   A COLUMNWISE LEAST-SQUARES FORMULATION

Write $E \triangleq \hat{W} - W - V$, whose $k$-th column is $e_k$. Then

$$\ell(\hat{W}) - C = \mathrm{tr}(E^\top H E) = \sum_{i,j=1}^{n} H_{ij} \langle e_i, e_j \rangle, \tag{33}$$

where $\langle \cdot, \cdot \rangle$ is the standard inner product on $\mathbb{R}^m$. To decouple columns, factor $H = LDL^\top$ with $L$ unit lower triangular and $D = \mathrm{diag}(d_1, \ldots, d_n) \succcurlyeq 0$ (pivoted $LDL^\top$ if needed). Setting $\widetilde{E} \triangleq L^\top E$ gives

$$\ell(\hat{W}) - C = \mathrm{tr}(\widetilde{E}^\top D \widetilde{E}) = \sum_{k=1}^{n} d_k \|\tilde{e}_k\|_2^2, \tag{34}$$

where $\tilde{e}_k$ is the $k$-th column of $\widetilde{E}$. Hence the $k$-th update is governed (up to the scalar weight $d_k$) by the Euclidean error of a columnwise target in the $L$-basis.

**Linearization of the quantizer.**   Around $z_k \triangleq W_k + r_{0,k} + \alpha v_k$, use a first-order/bias–noise model

$$Q(z_k) \approx z_k + b_k + \eta_k, \qquad \mathbb{E}[\eta_k] = 0, \quad \mathrm{Cov}(\eta_k) = \Sigma_{\eta,k}. \tag{35}$$

Then

$$\hat{W}_k - W_k \approx r_{0,k} + \alpha v_k + b_k + \eta_k \quad \implies \quad e_k \approx (r_{0,k} + \alpha v_k + b_k - v_k) + \eta_k. \tag{36}$$

For any deterministic vector $a$ and zero-mean $\eta$, $\mathbb{E}\big[\|a + \eta\|_H^2\big] = \|a\|_H^2 + \mathrm{tr}(H \Sigma_\eta)$, so the $\alpha$-dependent part of the loss reduces to an $H$-weighted least-squares problem.

Define the "ideal target"

$$t_k \triangleq v_k - r_{0,k} - b_k, \qquad \|x\|_H^2 \triangleq x^\top H x. \tag{37}$$

Discarding the $\alpha$-independent $\mathrm{tr}(H\Sigma_{\eta,k})$, the per-column objective is

$$J_k(\alpha) \triangleq \|\alpha v_k - t_k\|_H^2. \tag{38}$$

*General $H$.* The minimizer is

$$\alpha_k^\star = \frac{v_k^\top H t_k}{v_k^\top H v_k}, \tag{39}$$

*Decoupled ($LDL^\top$) basis.* Using equation 34, $J_k(\alpha) = d_k \|\alpha v_k - t_k\|_2^2$ so $d_k$ cancels and

$$\alpha_k^\star = \frac{v_k^\top t_k}{v_k^\top v_k}. \tag{40}$$

When quantization is (approximately) unbiased, $b_k \approx 0$, so $t_k \approx v_k - r_{0,k}$ and $\alpha_k^\star \approx 1 - \dfrac{v_k^\top r_{0,k}}{\|v_k\|_2^2}$.

### A.4.2   RIDGE-REGULARIZED ESTIMATOR

To control estimation noise, penalize the scalar gain:

$$J_k^{\mathrm{ridge}}(\alpha) = \|\alpha v_k - t_k\|_2^2 + \gamma_k \alpha^2, \tag{41}$$

yielding the closed form

$$\alpha_k(\gamma_k) = \frac{v_k^\top t_k}{v_k^\top v_k + \gamma_k} = (1 - \lambda_k) \alpha_k^\star, \quad \lambda_k \triangleq \frac{\gamma_k}{\gamma_k + \|v_k\|_2^2}. \tag{42}$$

Thus $\lambda_k \in [0, 1)$ is an explicit shrinkage factor. Layer-wise regularization is analogous with $v_k, t_k$ concatenated or summed.

### A.4.3   A VARIANCE-DRIVEN CHOICE OF $\gamma_k$

Let $H = LDL^\top$ with unit lower-triangular $L$ (so $a_k = u_k$ in the dual upper-triangular view and $r_{0,k} = (W_{:,k+1:n} - \hat{W}_{:,k+1:n}) L_{k+1:n,k}$). Two dominant noise sources motivate $\gamma_k$:

**(i) Propagated right-column noise.** Let the quantization error on column $j > k$ be $e_j$ with $\mathbb{E}[e_j] = 0$ and $\mathrm{Cov}(e_j) = s_j I_m$. Then

$$r_{0,k} = \sum_{j>k} e_j \, L_{jk} \Rightarrow \mathrm{Cov}(r_{0,k}) = \Big( \sum_{j>k} s_j \, L_{jk}^2 \Big) I_m. \tag{43}$$

Consequently,

$$\widehat{\alpha}_0 = 1 - \frac{v_k^\top r_{0,k}}{\|v_k\|_2^2}, \qquad \mathrm{Var}(\widehat{\alpha}_0) = \frac{\sum_{j>k} s_j \, L_{jk}^2}{\|v_k\|_2^2}. \tag{44}$$

A natural stabilizer is to *move* this propagated variance into the denominator:

$$\gamma_{r_0,k} \approx \sum_{j>k} s_j \, L_{jk}^2. \tag{45}$$

**(ii) Current-column noise floor.** Let $\sigma_{e,k}^2$ denote the per-dimension variance of the current column's quantization noise (estimated from recent residuals). Introduce a base ridge level $\gamma_0 \triangleq m \, \sigma_{e,k}^2$ to match dimensions.

**Combined ridge.** With observable residuals $e_{rj}$ on already-quantized columns $j > k$,

$$\gamma_k = \gamma_0 + \sum_{j>k} \Big( \sum_{r=1}^m e_{rj}^2 \Big) L_{jk}^2. \tag{46}$$

This choice yields a nearly hyperparameter-free $\alpha_k(\gamma_k)$ in equation 42 that adapts to both propagated and intrinsic noise.

## A.5 FULL RESULTS

### A.5.1 QUANTITATIVE RESULTS

In this section, we provide a comprehensive presentation of our results across various datasets to complement the main paper. The results include complete comparison of the perplexity score on WikiText2 and averaged accuracy on zero-shot common sense reasoning tasks on LLaMA-2(Tab 6), LLaMA-3 (Tab 7) and Qwen-3(Tab 8).

### A.5.2 SPEEDUP AND MEMORY SAVINGS

Tab 9 shows the prefill time and memory usage of LLaMA models with different parameter sizes and sequence lengths, compared between our W2A4 implementation and FP16. The inference environment features an Intel(R) Xeon(R) Gold 5317 CPU and an Nvidia 3090 GPU. The 4-bit matrix multiplication kernel was implemented using cutlass of nvidia, while the self-attention mechanism was realized with PyTorch's native SDPA (scaled dot product attention) function. All tests were conducted 500 times, with the median value taken as the final result. Benefiting from efficient low-precision computation units within CUDA cores and reduced access overhead, AEC-SVQ achieves over 3× speedup across various model sizes, and approximately 7× acceleration on the challenging LLaMA-30B model.

Table 6: Complete comparison of the perplexity score on WikiText2 and averaged accuracy on Zero-shot Common Sense Reasoning tasks on **LLaMA-2**.

| Model | Method | ARC-c (↑) | ARC-e (↑) | BoolQ (↑) | HellaS. (↑) | OBQA (↑) | PIQA (↑) | SIQA (↑) | WinoG. (↑) | Avg. (↑) | Wiki2 (↓) |
|---|---|---|---|---|---|---|---|---|---|---|---|
| 2-7B | Full Precision | 46.42 | 74.33 | 77.71 | 75.94 | 44.20 | 79.16 | 45.91 | 69.53 | 64.15 | 5.47 |
| | SmoothQuant | 23.29 | 26.52 | 46.18 | 26.16 | 21.80 | 47.77 | 33.37 | 50.20 | 34.41 | 6e5 |
| | OmniQuant | 23.72 | 25.11 | 37.95 | 26.27 | 23.80 | 48.20 | 34.39 | 50.04 | 33.69 | 4e5 |
| | QuaRot | 29.10 | 24.92 | 47.86 | 25.75 | 28.40 | 49.89 | 33.57 | 49.01 | 36.06 | 1e5.00 |
| | SpinQuant+GPTQ | 28.92 | 26.05 | 58.47 | 29.75 | 26.60 | 51.36 | 34.49 | 51.93 | 38.45 | 124.79 |
| | SpinQuant+GPTAQ | 29.35 | 26.09 | 61.65 | 29.27 | 27.00 | 50.22 | 34.85 | 51.07 | 38.69 | 7561.60 |
| | OSTQuant+GPTQ | 21.84 | 33.59 | 38.96 | 30.49 | 24.80 | 55.88 | 35.88 | 49.33 | 36.35 | 41.15 |
| | OSTQuant+GPTAQ | 24.91 | 34.13 | 62.29 | 39.36 | 30.20 | 57.34 | 36.80 | 54.70 | 42.47 | 12.46 |
| | **AEC-SVQ** | 35.67 | 62.29 | 67.65 | 67.31 | 38.40 | 74.65 | 41.45 | 62.19 | 56.20 | 6.29 |
| 2-13B | Full Precision | 49.15 | 77.53 | 80.58 | 79.39 | 45.20 | 80.63 | 47.49 | 71.90 | 66.48 | 4.88 |
| | SmoothQuant | 23.72 | 26.52 | 46.18 | 26.17 | 24.80 | 49.13 | 34.64 | 47.91 | 34.88 | 3e5 |
| | OmniQuant | 24.49 | 25.63 | 48.84 | 27.34 | 25.60 | 49.02 | 34.19 | 47.75 | 35.36 | 1e5 |
| | QuaRot | 27.22 | 25.93 | 51.35 | 26.52 | 27.40 | 50.05 | 34.14 | 47.59 | 36.28 | 9e4 |
| | SpinQuant+GPTQ | 24.83 | 39.56 | 61.47 | 38.01 | 27.00 | 54.84 | 35.62 | 53.51 | 41.85 | 23.64 |
| | SpinQuant+GPTAQ | 25.85 | 42.13 | 61.25 | 35.95 | 28.20 | 57.51 | 35.26 | 52.09 | 42.28 | 33.21 |
| | OSTQuant+GPTQ | 24.74 | 42.72 | 63.12 | 39.28 | 28.60 | 61.26 | 37.15 | 54.70 | 43.95 | 15.85 |
| | OSTQuant+GPTAQ | 28.41 | 48.95 | 63.46 | 44.26 | 32.40 | 63.82 | 37.67 | 55.72 | 46.84 | 8.90 |
| | **AEC-SVQ** | 41.72 | 68.06 | 74.37 | 73.41 | 41.60 | 76.93 | 43.91 | 66.22 | 60.78 | 5.49 |
| 2-70B | Full Precision | 57.42 | 81.02 | 83.79 | 83.81 | 48.80 | 82.70 | 49.18 | 77.98 | 70.59 | 3.32 |
| | SmoothQuant | 28.12 | 25.88 | 38.97 | 25.12 | 24.60 | 50.76 | 32.55 | 47.44 | 34.18 | 2e5 |
| | OmniQuant | 29.24 | 25.55 | 37.83 | 26.76 | 26.60 | 50.98 | 33.98 | 48.16 | 34.89 | 9e4 |
| | QuaRot | 28.92 | 27.40 | 37.92 | 25.65 | 23.00 | 50.00 | 33.78 | 46.65 | 34.17 | 8333.76 |
| | SpinQuant+GPTQ | 33.96 | 46.42 | 56.85 | 46.04 | 32.00 | 58.49 | 37.46 | 56.04 | 45.91 | 656.00 |
| | SpinQuant+GPTAQ | 36.77 | 62.84 | 61.47 | 50.19 | 36.40 | 70.51 | 38.18 | 64.17 | 52.57 | 200.00 |
| | OSTQuant+GPTQ | 32.08 | 46.42 | 60.83 | 55.64 | 35.60 | 62.19 | 40.74 | 66.38 | 49.99 | 11.31 |
| | OSTQuant+GPTAQ | 37.88 | 65.82 | 68.44 | 64.84 | 39.00 | 71.60 | 42.94 | 66.85 | 57.17 | 7.71 |
| | **AEC-SVQ** | 51.28 | 78.41 | 73.85 | 78.08 | 45.00 | 78.56 | 46.11 | 73.72 | 65.63 | 4.41 |

Table 7: Complete comparison of the perplexity score on WikiText2 and averaged accuracy on Zero-shot Common Sense Reasoning tasks on **LLaMA-3**.

| Model | Method | ARC-c (↑) | ARC-e (↑) | BoolQ (↑) | HellaS. (↑) | OBQA (↑) | PIQA (↑) | SIQA (↑) | WinoG. (↑) | Avg. (↑) | Wiki2 (↓) |
|---|---|---|---|---|---|---|---|---|---|---|---|
| 3-8B | Full Precision | 53.50 | 77.74 | 81.10 | 79.18 | 44.80 | 80.63 | 47.08 | 73.01 | 67.13 | 6.14 |
| | SmoothQuant | 22.24 | 24.28 | 48.78 | 26.40 | 25.40 | 50.33 | 34.14 | 50.20 | 35.22 | 2e6 |
| | OmniQuant | 25.77 | 24.07 | 56.27 | 25.68 | 26.60 | 50.60 | 32.12 | 50.59 | 36.46 | 2e6 |
| | QuaRot | 28.24 | 24.83 | 49.24 | 25.74 | 28.80 | 50.65 | 33.37 | 49.64 | 36.31 | 3e5 |
| | SpinQuant+GPTQ | 21.50 | 32.37 | 47.52 | 29.51 | 25.80 | 53.37 | 32.70 | 50.75 | 36.69 | 96.94 |
| | SpinQuant+GPTAQ | 22.78 | 36.74 | 59.57 | 34.24 | 26.20 | 55.71 | 34.85 | 52.88 | 40.37 | 48.31 |
| | OSTQuant+GPTQ | 21.93 | 34.01 | 50.43 | 31.62 | 26.40 | 55.82 | 35.16 | 51.30 | 38.33 | 36.20 |
| | OSTQuant+GPTAQ | 24.66 | 30.81 | 61.96 | 37.73 | 28.20 | 53.26 | 37.41 | 54.38 | 41.05 | 20.20 |
| | **AEC-SVQ** | 41.30 | 63.64 | 74.59 | 69.04 | 38.40 | 74.21 | 42.37 | 63.61 | 58.39 | 8.65 |
| 3-70B | Full Precision | 49.15 | 77.53 | 80.58 | 79.39 | 45.20 | 80.63 | 47.49 | 71.90 | 66.48 | 4.88 |
| | SmoothQuant | 27.47 | 26.05 | 37.83 | 26.26 | 24.80 | 50.98 | 32.46 | 48.38 | 34.28 | 7e5 |
| | OmniQuant | 24.15 | 25.88 | 37.83 | 26.12 | 26.40 | 50.76 | 32.55 | 49.17 | 34.11 | 6e5 |
| | QuaRot | 23.81 | 26.09 | 42.39 | 26.68 | 27.40 | 51.03 | 34.34 | 51.62 | 35.42 | 5e5 |
| | SpinQuant+GPTQ | 25.77 | 25.17 | 45.17 | 29.22 | 26.07 | 51.63 | 33.83 | 48.93 | 35.72 | 3e5 |
| | SpinQuant+GPTAQ | 27.13 | 25.29 | 48.81 | 32.48 | 26.12 | 51.74 | 34.12 | 48.70 | 36.80 | 4e5 |
| | OSTQuant+GPTQ | 26.37 | 27.27 | 54.56 | 33.33 | 28.40 | 51.96 | 32.60 | 52.17 | 38.33 | 618.90 |
| | OSTQuant+GPTAQ | 25.94 | 25.55 | 53.85 | 32.63 | 29.00 | 51.96 | 33.52 | 53.91 | 38.29 | 559.68 |
| | **AEC-SVQ** | 51.11 | 77.48 | 79.88 | 78.82 | 34.20 | 79.00 | 44.78 | 59.59 | 63.11 | 6.33 |

Table 8: Complete comparison of the perplexity score on WikiText2 and averaged accuracy on Zero-shot Common Sense Reasoning tasks on **Qwen-3**.

| Model | Method | ARC-c (↑) | ARC-e (↑) | BoolQ (↑) | HellaS. (↑) | OBQA (↑) | PIQA (↑) | SIQA (↑) | WinoG. (↑) | Avg. (↑) | Wiki2 (↓) |
|-------|--------|-----------|-----------|-----------|-------------|----------|----------|----------|------------|----------|-----------|
| | Full Precision | 56.23 | 80.93 | 86.67 | 74.91 | 41.40 | 78.07 | 51.84 | 68.19 | 67.28 | 9.72 |
| 3-8B | SmoothQuant | 26.62 | 23.57 | 38.01 | 26.20 | 29.60 | 51.58 | 33.32 | 50.91 | 34.98 | 282152.00 |
| | OmniQuant | 27.82 | 24.12 | 37.82 | 26.36 | 27.20 | 50.98 | 33.45 | 49.26 | 34.63 | 257368.00 |
| | QuaRot | 25.46 | 25.66 | 39.93 | 28.12 | 28.60 | 51.23 | 33.96 | 51.22 | 35.52 | 146378.00 |
| | SpinQuant+GPTQ | 27.30 | 43.60 | 65.32 | 42.20 | 28.40 | 60.01 | 33.62 | 52.80 | 44.16 | 24.57 |
| | SpinQuant+GPTAQ | 26.45 | 41.25 | 63.55 | 38.74 | 27.40 | 59.30 | 36.23 | 52.72 | 43.20 | 25.24 |
| | OSTQuant+GPTQ | 23.81 | 41.79 | 63.94 | 35.66 | 27.20 | 60.28 | 36.80 | 53.12 | 42.82 | 27.49 |
| | OSTQuant+GPTAQ | 29.18 | 46.34 | 69.08 | 44.69 | 27.60 | 61.04 | 38.08 | 52.69 | 46.12 | 24.62 |
| | **AEC-SVQ** | 46.93 | 71.46 | 78.38 | 66.21 | 39.40 | 72.58 | 44.11 | 65.11 | 60.52 | 11.27 |
| | Full Precision | 60.49 | 82.87 | 89.39 | 78.87 | 46.40 | 79.60 | 51.94 | 73.01 | 70.32 | 8.65 |
| 3-14B | SmoothQuant | 27.22 | 25.17 | 43.39 | 25.97 | 28.60 | 50.82 | 33.27 | 49.41 | 35.48 | 638472.00 |
| | OmniQuant | 27.04 | 26.77 | 46.20 | 25.50 | 29.20 | 51.73 | 32.36 | 51.02 | 36.23 | 537462.00 |
| | QuaRot | 28.33 | 24.62 | 50.76 | 26.45 | 28.00 | 52.07 | 33.42 | 52.64 | 37.04 | 263984.00 |
| | SpinQuant+GPTQ | 26.71 | 39.10 | 57.00 | 36.77 | 28.00 | 57.73 | 35.98 | 51.62 | 41.61 | 41.57 |
| | SpinQuant+GPTAQ | 28.41 | 44.61 | 72.81 | 44.82 | 29.80 | 63.06 | 39.20 | 57.22 | 47.49 | 17.04 |
| | OSTQuant+GPTQ | 35.67 | 59.18 | 76.67 | 50.70 | 34.40 | 66.05 | 40.17 | 60.38 | 52.90 | 17.55 |
| | OSTQuant+GPTAQ | 33.78 | 57.28 | 76.15 | 49.91 | 33.40 | 66.10 | 39.92 | 57.54 | 51.77 | 17.51 |
| | **AEC-SVQ** | 48.46 | 73.95 | 81.87 | 72.51 | 41.20 | 76.44 | 47.13 | 68.03 | 63.70 | 10.38 |

Table 9: Prefill time and Memory usage of LLaMA models with different parameter sizes and sequence lengths, compared between our 4-bit implementation and FP16. All tests were conducted on a Transformer block with batch size 4 on a 3090 GPU.

| Model Size | Seqlen | Prefill Time(ms) | | Prefill Speedup(×) | Memory(GB) | | Memory Saving(×) |
|------------|--------|------------------|--------|-------------------|------------|--------|------------------|
| | | FP16 | W2A4 | | FP16 | W2A4 | |
| | 256 | 8.050 | 3.326 | 2.420 | 0.411 | 0.066 | 6.266 |
| | 512 | 14.904 | 6.386 | 2.334 | 0.435 | 0.074 | 5.902 |
| | 1024 | 27.286 | 12.210 | 2.235 | 0.483 | 0.090 | 5.367 |
| LLaMA2-7B | 2048 | 54.979 | 24.720 | 2.224 | 0.577 | 0.122 | 4.728 |
| | 4096 | 112.603 | 51.020 | 2.207 | 0.766 | 0.187 | 4.103 |
| | 8192 | 224.275 | 129.630 | 1.730 | 1.147 | 0.317 | 3.615 |
| | 256 | 8.035 | 3.014 | 2.666 | 0.430 | 0.068 | 6.317 |
| | 512 | 15.545 | 6.036 | 2.575 | 0.442 | 0.073 | 6.051 |
| | 1024 | 29.169 | 11.128 | 2.621 | 0.466 | 0.083 | 5.613 |
| LLaMA3-8B | 2048 | 57.470 | 23.339 | 2.462 | 0.513 | 0.103 | 4.991 |
| | 4096 | 117.593 | 49.511 | 2.375 | 0.608 | 0.142 | 4.273 |
| | 8192 | 256.324 | 113.263 | 2.263 | 0.795 | 0.221 | 3.593 |
| | 256 | 11.449 | 4.080 | 2.806 | 0.634 | 0.095 | 6.686 |
| | 512 | 21.195 | 7.285 | 2.909 | 0.663 | 0.105 | 6.326 |
| | 1024 | 41.762 | 15.107 | 2.764 | 0.723 | 0.126 | 5.730 |
| LLaMA2-13B | 2048 | 81.955 | 31.936 | 2.566 | 0.841 | 0.165 | 5.096 |
| | 4096 | 199.046 | 69.881 | 2.848 | 1.079 | 0.247 | 4.372 |
| | 8192 | 359.402 | 154.080 | 2.333 | 1.553 | 0.409 | 3.799 |
| | 256 | 18.689 | 5.174 | 3.612 | 1.047 | 0.148 | 7.082 |
| | 512 | 34.393 | 10.824 | 3.177 | 1.085 | 0.162 | 6.699 |
| | 1024 | 66.880 | 21.902 | 3.054 | 1.162 | 0.187 | 6.197 |
| LLaMA-30B | 2048 | 157.585 | 45.680 | 3.450 | 1.315 | 0.240 | 5.493 |
| | 4096 | 272.355 | 95.229 | 2.860 | 1.625 | 0.346 | 4.697 |
| | 8192 | 576.555 | 214.940 | 2.682 | 2.242 | 0.557 | 4.029 |

