# OpenReview forum: "Towards W2A4 LLM Inference: Hybrid SQ-VQ Framework with Adaptive Error Compensation"
_ICLR.cc/2026/Conference — Submitted to ICLR 2026_

### Official Review · Reviewer_QyPF · 2025-10-27

**Soundness:** 3
**Presentation:** 2
**Contribution:** 3
**Rating:** 4
**Confidence:** 2

**Summary:**

This paper proposes AEC-SVQ, a quantization framework for large language models that employs scalar quantization for activations and vector quantization for weights, utilizing INT4 arithmetic units. It introduces an effective transformation for quantization, a novel quantization algorithm that accounts for cumulative errors across quantized layers, and adaptive correction methods to further enhance accuracy.

**Strengths:**

* The proposal to synergistically use scalar quantization for activations and vector quantization for weights, while also quantizing the codebook to leverage low-precision ALUs, is brilliant and makes this work highly relevant to real-world practitioners.
* AEC-SVQ is extensively compared against several existing solutions and demonstrates promising result.

**Weaknesses:**

* The overall contributions of this work are somewhat difficult to grasp at a high level.
* The speedup evaluations are incomplete.

**Questions:**

* First, I would like to ask about two of the three main contributions claimed in the paper: learned transformation and CEAVQ.

Regarding the learned transformation, I am curious how the proposed mechanism differs from existing learned rotation or transformation approaches such as SpinQuant or DuQuant, and why it performs better both quantitatively and qualitatively.

As for CEAVQ, I found it somewhat difficult to understand the core novelty that distinguishes it from prior work, partly due to my lack of mathematical literacy. To me, the objective function in Equation (5) does not appear particularly novel, and it seems that the key contribution lies in Equation (6) and the accompanying discussion. I would greatly appreciate a higher-level explanation of CEAVQ’s conceptual contribution beyond the mathematical formulation.

Minor: Why not just write equation (5) as W^X-WX~? This seems more intuitive to me.

* Second, I am wondering why decode speedup results are not included in the manuscript. I assume that this is because achieving decode speedup would require a specialized kernel supporting vector quantization with quantized codebooks. If that is the case, I understand the challenge. However, including at least a discussion on this limitation or providing partial evidence of end-to-end speedup would make the paper more valuable.

* Third, low-precision floating-point formats (e.g., FP4) are becoming increasingly popular for LLM inference as newer GPUs offer native support. How would the proposed approach extend to support FP4 instead of INT4? What would you expect in terms of performance and accuracy? Would it outperform the INT4 version, or not?

---

> ### Author Response · Authors · 2025-11-21
> **Rebuttal for Reviewer QyPF. Part [1]**
>
> **Dear Reviewer QyPF,**
>
> We sincerely thank you for your thorough evaluation and constructive comments. We address your valuable comments point by point as follows:
>
> ---
>
> > The overall contributions of this work are somewhat difficult to grasp at a high level.
>
> We thank the reviewer for this high-level feedback. To clarify, our work addresses the challenge of **enabling efficient W2A4 inference (2-bit weights and 4-bit activations) by systematically coupling Vector Quantization (VQ) and Scalar Quantization (SQ)**. The overall narrative and core contributions are structured as follows:
>
> ### **1. Core Goal and Architecture: The Hybrid SQ-VQ Framework**
>
> - **Objective:** Our primary goal is to simultaneously achieve the memory compression benefits of 2-bit weights and the computational acceleration of 4-bit integer matrix multiplication (INT4 GEMM).
> - **Architecture:** To satisfy the distinct properties of weights and activations, we propose a Hybrid SQ-VQ architecture. This design leverages VQ for weights to maximize compression and SQ for activations to maintain hardware compatibility.
>
> ### **2.1 Addressing Quantization Difficulty: Learnable Transformation**
>
> - **Challenge:** Direct quantization of weights and activations to such low bit-widths is difficult due to irregular distributions and outliers.
> - **Innovation:** We introduce a unified Learnable Transformation ($T = \Lambda O$). Unlike standard outlier suppression techniques, this transformation serves a triple purpose:
>     - It conditions the weights in the transformed domain to be more amenable to VQ.
>     - It conditions the activations in the inverse-transformed domain to be suitable for SQ.
>     - It aligns the quantized codebook more naturally with the INT4 grid required by hardware.
>
> ### **2.2 Addressing Cross-layer Cumulative**
>
> - **Challenge:** Even after distributions are made more quantization-friendly, **ultra-low-bit quantization still induces large errors that accumulate layer by layer.** Existing PTQ/VQ methods typically minimize a local reconstruction objective per layer, fail to account for how errors accumulate across layers.
> - **Innovation:** We propose **Cumulative-Error-Aware VQ (CEAVQ)**. Instead of minimizing local reconstruction error for a single layer, we derive a global proxy that explicitly models both the **current weight quantization error** and the **accumulated upstream weight-activation quantization errors**. This results in a pre-compensated weight target $W_{\text{opt}} = W + WGH^{-1}$ that proactively aligning the layer's output with the full-precision output distribution.
>
> ### **2.3 Robustness Solution: Adaptive Compensation under limited calibration**
>
> - **Challenge:** The theoretical compensation derived in CEAVQ relies on Hessian and statistical estimates. **Under limited calibration data, applying this compensation directly entails a high risk of overfitting**.
> - **Innovation:** To resolve this, we introduce **Adaptive Compensation via Bias-Variance Shrinkage**. We formalize the compensation as a regularization problem involving a bias-variance tradeoff. This leads to a **closed-form column-wise shrinkage solution** $\hat W_k = W_k + \alpha_k(WGH^{-1})_k$, where $\alpha_k$ adaptively adjusts the compensation strength based on the noise level of each column. This ensures the method generalizes well to unseen data, **which is critical for the stability of W2A4 inference**.
>
>
> ### **3 Empirical evidence: SOTA performance in extreme W2A4 and 2-bit weight-only regimes**
> Our system-level experiments further support that AEC-SVQ is more than a simple stacking of existing modules:
> - Under the **W2A4 setting**, we compare against strong SQ-based baselines such as SmoothQuant, OmniQuant, QuaRot, SpinQuant, and OSTQuant (**Table 1**). AEC-SVQ achieves **SOTA** results on WikiText2 perplexity and multiple zero-shot tasks.
> - Under the **2-bit weight-only setting**, we compare against recent VQ-based methods such as QuIP, QuIP#, PCDVQ, and VPTQ (**Table 4**), and obtain **SOTA** performance under the same backbones and settings.
>
> These results indicate that AEC-SVQ is not a superficial combination of two existing lines, but a coherent framework designed around a unified objective, with theoretical backing and practical deployability on INT4 hardware.

---

> > ### Author Response · Authors · 2025-11-21
> > **Rebuttal for Reviewer QyPF. Part [2]**
> >
> > ### **4 Summary of innovations**
> > In summary, AEC-SVQ is not a simple combination of existing VQ and SQ techniques.
> > Instead, aiming to realize  practical and high-accuracy W2A4 LLM inference, we systematically introduce:
> >   - A **unified hybrid SQ–VQ framework** with a single learnable transform $T$ that simultaneously shapes weights, activations, and INT4 codebook structure under a joint W2A4 objective.
> >   - **CEAVQ**, which moves from local, single-layer proxies to a **cumulative-error-aware VQ** formulation with a closed-form optimal pre-compensated weight $W_{\text{opt}} = W + W G H^{-1}$ and a column-wise, feedback-aware VQ procedure.
> >   - A **column-wise adaptive shrinkage mechanism** for compensation derived from the same cumulative-error proxy, addressing bias–variance trade-offs **under limited calibration data.**
> >   - **Systematic SOTA results** in both W2A4 and 2-bit weight-only regimes, demonstrating that these innovations yield practical gains beyond a simple combination of existing SQ and VQ techniques.
> >
> > > The speedup evaluations are incomplete.
> >
> > > Second, I am wondering why decode speedup results are not included in the manuscript. I assume that this is because achieving decode speedup would require a specialized kernel supporting vector quantization with quantized codebooks. If that is the case, I understand the challenge. However, including at least a discussion on this limitation or providing partial evidence of end-to-end speedup would make the paper more valuable.
> >
> > We thank the reviewer for this valuable insight regarding practical deployment. We agree that decode latency is a critical metric for LLM inference. We address your concern regarding the speedup evaluations from three aspects:
> >
> > ### **1 Existing Evaluations on Prefill and Memory**
> > - In the current manuscript, we have explicitly reported the prefill speedup and memory savings of AEC-SVQ (W2A4) compared to FP16 in **Table 2 of the main paper and Table 9 in Appendix A.5.2**
> > - Notably, for the LLaMA-30B model, we achieved approximately $3\times$ speedup in the prefill phase and $7\times$ reduction in memory usage. We initially prioritized the prefill phase because it often constitutes the primary throughput bottleneck for large models processing long sequences.
> >
> > ### **2 Analysis of Decode Latency**
> > - Regarding the decoding phase, it is fundamental to note that token generation in LLMs is typically **memory-bandwidth bound** rather than compute bound.
> > - Consequently, the decoding speed is primarily determined by **the storage volume of the weights and the memory access bandwidth**.
> > - Since AEC-SVQ compresses weights to 2 bits, our theoretical decoding speedup is **comparable to that of other SOTA 2-bit weight quantization methods**, but with the **additional benefit of INT4 compute**.
> >
> > ### **3 Supplementary Experiments on Decode Speedup**
> >
> > To further address your concern, we have conducted additional experiments to specifically measure the decoding speed on NVIDIA A6000 GPU.
> >
> > As shown in **Table R4-1**, AEC-SVQ achieves significant speedups in the decoding phase, confirming that our method delivers consistent efficiency gains across both prefill and decode stages.
> >
> > **Table R4-1: Comparison of generation speed in the decoding stage.**
> > | Model       | Model Decoder Speed (tokens/sec) |        | Speed up |
> > |:-----------:|:--------------------------------:|:------:|:--------:|
> > |             | FP                               | W2A4   |          |
> > | LLaMA-2-7B  | 45.86                            | 140.79 | 3.07x    |
> > | LLaMA-3-8B  | 36.02                            | 114.54 | 3.18x    |
> > | LLaMA-2-13B | 24.3                             | 80.43  | 3.31x    |

---

> > > ### Author Response · Authors · 2025-11-21
> > > **Rebuttal for Reviewer QyPF. Part [3]**
> > >
> > > > Regarding the learned transformation, I am curious how the proposed mechanism differs from existing learned rotation or transformation approaches such as SpinQuant or DuQuant, and why it performs better both quantitatively and qualitatively.
> > >
> > > We thank the reviewer for this specific inquiry regarding the learned transformation and for drawing connections to related works. Clarifying the distinction between our approach and these methods is indeed central to understanding the positioning of our paper.
> > >
> > > **The fundamental distinction lies in the scope of the transformation.** Our transformation is not merely a rotation designed solely for the scalar quantization of activations like SpinQuant and DuQuant. Instead, our transformation $T$ to is systematically designed to serve three distinct objectives simultaneously within a hybrid framework:
> > > - Conditioning weights in the transformed space to be more **suitable for Vector Quantization (VQ) by making the covariance more balanced and geometrically isotropic**.
> > > - Conditioning activations in the transformed space to have a smaller dynamic range, making them **suitable for high-precision Scalar Quantization** (SQ).
> > > - Ensuring that the codebook generated by VQ is inherently **suitable for direct mapping onto the 4-bit integer grid required for GEMM operations**.
> > >
> > > We detail these differences in terms of the **target scope** and the **optimization objectives**, followed by an **explanation** of the resulting quantitative and qualitative superiority.
> > >
> > > ### **(1) Difference in Target Scope**
> > > - **Existing Methods:** Approaches like SpinQuant, OSTQuant, and DuQuant focus primarily on **mitigating outliers in the activation distribution to minimize the error of activation scalar quantization.** These works generally do not explicitly model the covariance structure required for weight Vector Quantization, the structural constraints of the codebook/INT4 grid, or the unified trade-off between activation SQ, weight VQ, and 4-bit operator compatibility.
> > > - **Our Approach:** In contrast, the transformation $T$ in our work is designed under the unified proxy defined in Eq. (5)/(10). Its target is consistently the holistic "Hybrid SQ–VQ for W2A4 inference" problem.
> > >     - In the domain of $T$, we aim for **weight covariance to approach isotropy**, which renders VQ and codebook learning more efficient and stable.
> > >     - In the domain of $T^{-1}$, we aim to **compress the dynamic range of activations and shorten their tails**, making them more suitable for 4-bit SQ.
> > >     - Simultaneously, the weight codebook is constrained to fall directly onto a 4-bit computation-friendly integer grid, ensuring 4-bit compute efficiency.
> > >
> > > Consequently, our transformation $T$ does not operate as a preliminary rotation followed by decoupled weight and activation quantization. Instead, it is explicitly designed as a unified mechanism that **simultaneously optimizes weight VQ, activation SQ, and 4-bit codebook compatibility.** This holistic integration constitutes a **fundamental distinction from prior approaches such as SpinQuant and DuQuant**.
> > >
> > > ### **(2) Difference in Optimization Objectives**
> > > - **Existing Methods:** The training objectives of SpinQuant or OSTQuant are biased towards **reducing the local error of activation SQ**. They do not explicitly account for the reconstruction error of weight VQ, the error from codebook integer quantization, or the joint impact of these factors in a W2A4 scenario.
> > > - **Our Approach:** In Appendix A.2, we first construct **a unified error proxy that incorporates Weight VQ error, Activation SQ error, and Codebook integer quantization error into a single formulation** (Eq. 10). We then prove that, under given assumptions, there **exists a class of transformations $T^\star = H^{-1/2} O^\star$ that strictly reduces this unified error relative to the identity case $T=I$.** Therefore, our learned $T$ is not just a rotation for activation outlier suppression but is aligned with this unified proxy. The resulting $T$ balances Weight VQ, Activation SQ, and the 4-bit grid constraints, rather than optimizing for a single side.

---

> > > > ### Author Response · Authors · 2025-11-21
> > > > **Rebuttal for Reviewer QyPF. Part [4]**
> > > >
> > > > ### **(3) Superiority in Quantitative and Qualitative Performance**
> > > >
> > > > - **Quantitative Superiority in Stringent W2A4 Scenarios**
> > > >
> > > >   Our experimental results demonstrate that this unified design yields substantial gains in extremely low-bit settings:
> > > >     - If one uses **only a rotation + SQ approach** similar to SpinQuant in a W2A4 setting, the performance degrades rapidly. This is because while activation outliers are reshaped, the weights are still treated with extreme-low-bit SQ.
> > > >     - If one uses **only a VQ scheme like QuIP# or PCDVQ** without unified transformation and 4-bit activation reshaping, the activation outliers and cumulative error in the W2A4 scenario becomes excessive, leading to significant deterioration in performance.
> > > >     - **AEC-SVQ:** By employing the $T$ learned under our unified proxy, we **simultaneously suppress both weight VQ error and activation SQ error.** This leads to performance that is significantly superior to the aforementioned baselines in both W2A4 and 2-bit weight-only scenarios (see **Table 1** and **Table 4**).
> > > >
> > > >     These results substantiate that our contribution extends beyond minor variations in rotation implementation details. Instead, **it constitutes a fundamental advancement in both objective formulation and synergistic co-design**.
> > > >
> > > > - **Qualitative Distribution Reshaping**
> > > >
> > > >   The visualization of distributions (Figure 2) further highlights the difference.
> > > >     - In the $T$ domain, the weights achieve greater **spectral isotropy and covariance balance**, and the codeword distribution becomes more concentrated, facilitating compact representation for VQ.
> > > >     - In the $T^{-1}$ domain, he activation histograms demonstrate **significant tail shrinkage and outlier reduction**, which creates a favorable condition for 4-bit SQ.
> > > >     - This phenomenon of **simultaneous bilateral enhancement via a single transformation** is absent in former rotation methods tailored exclusively for activation SQ.
> > > >
> > > > In conclusion, we emphasize that the learnable transformation $T$ in this work is not a local technique for rotating single-sided scalar quantization. **It is systematically designed around the unified error proxy presented in Eq. (5)/(10) to serve weight VQ, activation SQ, and the realizability of codebooks on 4-bit integer cores simultaneously.** Furthermore, driven by this unified proxy theory, we prove **the existence of a transformation class $T^\star = H^{-1/2}O^\star$ that strictly reduces the unified error relative to $T=I$, achieving quantitative improvements and better distribution reshaping than representative methods like SpinQuant and QuIP# under extreme W2A4 settings.** We believe this demonstrates that our learned transformation offers an essential extension over existing work in terms of optimization objectives and target scope.

---

> > > > > ### Author Response · Authors · 2025-11-21
> > > > > **Rebuttal for Reviewer QyPF. Part [5]**
> > > > >
> > > > > > As for CEAVQ, I found it somewhat difficult to understand the core novelty that distinguishes it from prior work, partly due to my lack of mathematical literacy. To me, the objective function in Equation (5) does not appear particularly novel, and it seems that the key contribution lies in Equation (6) and the accompanying discussion. I would greatly appreciate a higher-level explanation of CEAVQ’s conceptual contribution beyond the mathematical formulation.
> > > > >
> > > > > We appreciate the reviewer for their candid feedback regarding the conceptual clarity of CEAVQ. We understand that mathematical formulations can sometimes obscure the underlying intuition. Here, we provide a high-level explanation to clarify the **specific problem CEAVQ addresses**, **its core idea**, and **how it distinguishes itself from existing methods** like GPTQ or standard VQ.
> > > > >
> > > > > ### **(1) Problem: Two Distinct Sources of Error**
> > > > > CEAVQ is motivated by a specific difficulty of ultra-low-bit W2A4 inference: in this regime, there are two coupled sources of error that jointly determine a layer’s output quality.
> > > > > - **Local Weight Error:** The error introduced within the current layer when compressing floating-point weights to VQ codes.
> > > > > - **Cumulative Upstream Error:** The error propagated from previous layers and the activation quantization itself (compressing high-precision activations to 4-bit). These errors accumulate and distort the input features received by the current layer.
> > > > >
> > > > > These two errors interact and can amplify each other as they propagate through depth, becoming the main bottleneck in W2A4 (as shown in **Fig.3**).
> > > > >
> > > > > ### **(2) The Core Idea: Active Error Cancellation**
> > > > > - **Most prior methods**, including GPTQ-style PTQ and existing VQ approaches focus primarily on the first source. Their goal is to minimize the **Local Reconstruction Error** under the assumption that its **input distribution is the one observed in full precision.** In other words, they focus on “make $\hat W$  close to $W$ for the current inputs.” This is effective at higher bitwidths, but in W2A4 the **input distribution has already drifted**, so minimizing only the local weight error is no longer sufficient.
> > > > > - The **fundamental philosophy of CEAVQ** is to select a quantized weight $\hat{W}$ such that these **two error sources mutually offset each other at the output, rather than merely minimizing the local weight quantization error in isolation**.It formalizes a different optimization goal for ultra-low-bit inference: align each layer to the full-precision reference under the already-quantized inputs, so that the layer proactively compensates for upstream drift.
> > > > >
> > > > >
> > > > > ### **(3) Deconstructing the Algorithm Conceptually**
> > > > > Algorithmically, the update rule in **Eq.6** can be conceptually divided into two distinct components:
> > > > > - **Component 1: Classical Residual Feedback**
> > > > >     - This term corresponds to the standard column-wise **residual feedback**.
> > > > >     - **Mechanism:** As we quantize columns sequentially, earlier columns introduce weight errors. We adjust the subsequent columns to compensate for these specific weight discretization errors.
> > > > >     - Goal: This strictly targets the **minimization of local weight reconstruction error**.
> > > > >
> > > > > - **Component 2: Cumulative Error Compensation (The Novel CEAVQ Component)**
> > > > >
> > > > >     This component represents the novel contribution of the CEAVQ framework. It explicitly leverages two key statistical properties:
> > > > >     - $H$: The covariance structure of the current layer inputs.
> > > > >     - $G$: The correlation between the accumulated upstream quantization error and the current input signal.
> > > > >     - The term $GH^{-1}$ mathematically defines an optimal adjustment vector. Modulating the weights along this specific trajectory maximizes the cancellation of propagated errors.
> > > > >     - Consequently, $W G H^{-1}$ addresses a fundamental optimization objective. It determines how to fine-tune the weights of the current layer to actively counteract the accumulated biases resulting from upstream quantization and activation compression, all while preserving the original functional integrity of the network.
> > > > >
> > > > > ### **Summary of Conceptual Contribution**
> > > > > Ultimately, these two components are combined to form the target vector for the VQ codebook projection. This vector simultaneously incorporates local compensation for weight quantization and global compensation for upstream cumulative error.From this perspective, **the conceptual contribution of CEAVQ is that it elevates vector quantization from a mechanism of "local reconstruction only" to a mechanism of "explicit network-level cumulative error compensation."** It achieves this by algorithmically integrating classical residual feedback with our proposed cumulative error correction.

---

> > > > > > ### Author Response · Authors · 2025-11-21
> > > > > > **Rebuttal for Reviewer QyPF. Part [6]**
> > > > > >
> > > > > > > Minor: Why not just write equation (5) as W^X-WX~? This seems more intuitive to me.
> > > > > >
> > > > > > We thank the reviewer for this helpful suggestion regarding notation. We confirm that your proposed formulation $\hat W X - W \tilde X$ is mathematically equivalent to our original expression, as demonstrated below:$$(\hat W - W)X - W(\tilde X - X) = \hat W X - WX - (W\tilde X - WX) = \hat W X - W\tilde X$$We initially selected the expanded form for the following reasons:
> > > > > > - **Explicit Decomposition**: It explicitly separates the two distinct sources of error. specifically the weight quantization error $(\hat W - W)X$ and the activation quantization error $W(\tilde X - X)$.
> > > > > > - **Derivation Clarity:** This decomposition facilitates the subsequent derivation of the block-wise form of CEAVQ and highlights its theoretical connection to the residual feedback mechanism.
> > > > > >
> > > > > > However, we fully concur with your observation that the form $\hat W X - W \tilde X$ offers superior intuitive clarity for the reader. Consequently, we have adopted your suggested notation in the revised manuscript.
> > > > > >
> > > > > > > Third, low-precision floating-point formats (e.g., FP4) are becoming increasingly popular for LLM inference as newer GPUs offer native support. How would the proposed approach extend to support FP4 instead of INT4? What would you expect in terms of performance and accuracy? Would it outperform the INT4 version, or not?
> > > > > >
> > > > > > We fully concur with the reviewer regarding the growing importance of low-precision floating-point formats (such as FP4 and NVFP4) in the context of next-generation GPU architectures. We address your questions regarding **extensibility**, **accuracy**, and **performance** as follows:
> > > > > >
> > > > > > **(1) Extensibility: Format-Agnostic Design**
> > > > > > - **Seamless Integration:** AEC-SVQ is inherently format-agnostic. From a methodological standpoint, our core framework relies only on the existence of a 4-bit discrete representation space. Consequently, extending the approach to support FP4 or NVFP4 is conceptually straightforward and requires replacing the INT4 quantizer with an FP4 quantizer.
> > > > > > - **Implementation:** In our framework, both SQ and VQ map weights and activations to a finite set $C$. To support FP4, we simply define $C$ as the non-uniform floating-point grid corresponding to the hardware standard.
> > > > > > - **No Migration Cost:** Critical modules such as CEAVQ and Adaptive Compensation operate on weight and activation statistics within the continuous domain. They are independent of the specific shape of the final encoding grid. Therefore, the migration to FP4 involves negligible algorithmic overhead.
> > > > > >
> > > > > > **(2) Expected Accuracy: FP4 $\ge$ INT4**
> > > > > >
> > > > > > We anticipate that the FP4 version would likely outperform the INT4 version in terms of accuracy.
> > > > > > - **Distribution Alignment:** FP4 typically offers a wider dynamic range and a logarithmic distribution. This aligns better with the approximate Gaussian or light-tailed distributions often observed in LLM weights and activations after rotation, a property supported by findings in works like OSTQuant and SpinQuant.
> > > > > > - **Outlier Management:** For layers exhibiting significant outliers, FP4 accommodates extreme values more naturally without excessive reliance on scaling factors.
> > > > > > - **Conclusion:** Therefore, under an identical 4-bit budget, we expect the FP4 implementation to yield lower quantization error and superior downstream task performance compared to INT4.
> > > > > >
> > > > > > **(3) Expected Performance: Comparable or Superior**
> > > > > >
> > > > > > We expect the inference speed to be comparable to, or slightly faster than, the INT4 version.
> > > > > > - **Throughput:** On devices offering native FP4 Tensor Cores, the computational throughput is generally comparable to that of INT4.
> > > > > > - **Conversion Overhead:** FP4 may offer slight advantages in quantization and dequantization overhead, as floating-point to floating-point conversions are often faster and simpler than floating-point to integer conversions.
> > > > > > - **Latency:** Since our framework is computationally format-agnostic, as long as the underlying 4-bit MAC throughput is comparable, the overall end-to-end inference latency will remain similar.
> > > > > >
> > > > > >
> > > > > > In conclusion, we expect the proposed approach to extend naturally to FP4. In terms of performance and accuracy, we predict that the FP4 version will achieve accuracy equal to or greater than the INT4 version (especially for outlier-heavy models) while maintaining equivalent inference speeds. We will explicitly highlight this extension as a promising direction for future work in the revised manuscript.

---

> ### Comment · Reviewer_QyPF · 2025-11-25
>
> Thank you for the detailed response to my comments.
> Your explanations have mostly resolved my concerns.
>
> I would like to request a quick clarification.
> Did you implement custom kernels for W2A4 in the decode-latency evaluation? If so, could you share a bit more detail about the implementation? I would also appreciate any insight into why the observed decode speedup does not match the theoretical speedup (~8×).

---

> > ### Author Response · Authors · 2025-11-27
> > **Gentle Follow-up in the Final Week of Discussion**
> >
> > **Dear Reviewer QyPF,**
> >
> > Thank you again for your careful reading of our work and for the constructive, insightful comments you provided. As we are now entering the final week of the discussion period, we would like to kindly follow up to see whether you have had the opportunity to review our updates.
> >
> > Following your suggestions, we have provided:
> >  - **Custom Kernel Details**: A breakdown of our W2A4 Triton implementation, including the On-the-fly Decode pipeline and memory hierarchy optimizations.
> >  - **Speedup Analysis**: An explanation of why the observed speedup (~3.6x) diverges from the theoretical limit due to non-linear overheads and VQ decoding complexity.
> >  - **Comparative Performance**: Benchmarks demonstrating that AEC-SVQ achieves superior decoding speedup compared to SOTA 2-bit methods (QTIP, VPTQ) while maintaining significant memory savings.
> >
> > We completely understand that you may have many commitments, and we are truly grateful for any time you could spare. If there is anything further that would help clarify our work, we would be more than happy to provide it. We respectfully hope you might reconsider the evaluation of our work.
> >
> > Thank you again for your thoughtful evaluation and valuable insights.
> >
> > Best regards,
> >
> > The Authors

---

> ### Author Response · Authors · 2025-11-26
> **Clarification on W2A4 Kernel Implementation and Decode Speedup Analysis. Part [1]**
>
> **Dear Reviewer QyPF,**
>
> **We sincerely thank you for the feedback and are glad that our previous responses have resolved most of your concerns.** Regarding your request for clarification on the W2A4 implementation and the observed speedup, we provide a detailed explanation below.
>
> ### **1 Custom Kernel Implementation Details**
>
> **Yes, we implemented custom kernels for the W2A4 pipeline.** To ensure maximum efficiency and flexibility, we developed our kernels using Triton. These kernels are utilized in both the prefill and decode phases.
>
> The **computation pipeline** is designed as follows:
>
> * **Stage 1: Pre-processing (Online Activation Quantization).** Input activations are quantized online to INT4. This involves calculating dynamic scaling factors (min/max) per token before the matrix multiplication.
> * **Stage 2: Core Computation (W2A4 VQ-GEMM).** We perform **On-the-fly Decode** within the kernel. The kernel loads compressed indices and INT4 codebooks, decodes them into weights in the registers, and immediately feeds them into Tensor Cores for accumulation into INT32.
> * **Stage 3: Post-processing.** The INT32 accumulators are dequantized back to FP16/BF16 using the combined scaling factors ($Scale_A \times Scale_W$).
>
> **Key Design Optimizations:**
> * **Data Layout & Memory Hierarchy:**
>     * **Codebook:** Given the compact size of our codebook ($4096 \times 6$ entries at 4-bit precision, totaling 12 KB), we explicitly designed the kernel to ensure it remains resident in Shared Memory. This strategy facilitates efficient codebook reuse and reduces latency overhead associated with global memory accesses.
>     * **Indices:** Indices are stored in a compressed format, where two 12-bit indices are packed into 3 bytes, to maximize memory bandwidth utilization.
> * **Tiling Strategy:** We employ a specialized block tiling strategy to handle the vector length $v=6$. Specifically, since the Vector Quantization is applied along the Output Channel (N) dimension, the block size $N$ necessitates alignment with $v$. We configured the Block Tile as $M=32$($M=1$ for decode), $N=96, K=64$, where $N=96$ is chosen to align with the vector length (96 \% 6 == 0) while maintaining sufficient occupancy for the GPU.
>
> ### **2 Analysis: Why Observed Speedup < Theoretical Speedup**
>
> You correctly noted that the theoretical compression of 2-bit weights suggests an 8x reduction in memory traffic compared to FP16. However, the observed end-to-end speedup is approximately **3.6x**. This discrepancy is expected and stems from two primary factors where practical hardware constraints diverge from theoretical ideals:
>
> **2.1 Non-Linear Overheads**
>
> Theoretical speedups often assume that the Matrix Multiply (GEMM) accounts for 100% of the latency. In reality, our W2A4 pipeline introduces necessary overheads:
> * **Online Quantization Cost:** Unlike static weight quantization, activations must be quantized **online** at runtime. This adds a computational cost (min/max reduction and scaling) to every layer that is not accelerated by INT4 Tensor Cores.
> * **Non-Quantized Operators:** Essential components like LayerNorm, MultiHeadAttention remain in Full precision. As the GEMM latency decreases, these fixed-cost operations dominate the total inference time, capping the maximum achievable speedup.
>
> **2.2 VQ Decoding Complexity**
>
> W2A4 inference is not a simple "Load $\rightarrow$ Compute" operation but a complex **"Load $\rightarrow$ Lookup $\rightarrow$ Compute"** process.
> * **Indirect Memory Access:** Decoding involves gathering INT4 values from the codebook using indices. Although the codebook is in Shared Memory, this indirect addressing introduces latency and register pressure that standard linear quantization lacks.
> * **Misalignment Overhead ($v=6$):** Tensor Cores is optimized for power-of-2 dimensions (e.g., 32, 64). Our vector length of $v=6$ necessitates complex pointer arithmetic and padding logic within the kernel to align data for the Tensor Core, reducing the instruction pipeline efficiency.

---

> ### Author Response · Authors · 2025-11-26
> **Clarification on W2A4 Kernel Implementation and Decode Speedup Analysis. Part [2]**
>
> ### **3 Comparative Analysis: Superior Speed Up & Significant Memory Savings**
>
> Despite the aforementioned constraints on latency, it is crucial to highlight that AEC-SVQ achieves near-theoretical gains in memory efficiency while still delivering state-of-the-art decoding speed.
>
> **3.1 Speed Comparison vs. SOTA**
>
> We benchmarked AEC-SVQ against 2-bit quantization frameworks on LLaMA-2-7B:
>
> **Table R4-2: Comparison of decoding speed between different methods.**
> |Method|Precision Scheme|Speed (tokens/sec)|Speedup (Ours vs. Others)|
> |:-:|:-:|:-:|:-:|
> |AEC-SVQ (Ours)| W2A4 (4-bit compute)|140.79|-|
> |QTIP|2-bit Weight-Only|116|1.21x|
> |VPTQ|2-bit Weight-Only|39.9|3.53x|
>
> As shown in **Table R4-2**, our W2A4 kernel achieves superior decoding throughput. While the ~3.6x speedup falls short of the theoretical limit due to the overheads mentioned above, our optimized W2A4 kernel achieves state-of-the-art decoding speed, outperforming existing VQ solutions. Notably, our design features a **highly compact codebook that is 4$\times$ smaller than that of VPTQ** due to INT4 codebook, offering a distinct advantage in memory efficiency and cache utilization.
>
>
> **3.2 Substantial Memory Footprint Reduction**
>
> While the latency speedup is ~3.6x, our method realizes the full potential of 2-bit compression in terms of capacity. AEC-SVQ achieves a memory saving factor of up to 7.1x (e.g., on LLaMA-30B) compared to FP16. This massive reduction **allows significantly larger models to be deployed on memory-constrained consumer GPUs**, addressing the **primary bottleneck for edge deployment where memory capacity is often more critical than latency**.
>
>
> We hope this detailed breakdown clarifies the implementation and the rationale behind the performance metrics. **Given these clarifications, we respectfully hope you might reconsider the evaluation of our work.**

---

### Official Review · Reviewer_HrU8 · 2025-10-31

**Soundness:** 2
**Presentation:** 2
**Contribution:** 2
**Rating:** 4
**Confidence:** 4

**Summary:**

The paper proposes to combine scalar quantization and vector quantization. Essentially the proposal is to use vector quantization on weights, but then quantize the entries inside the codebooks to INT4 representation. Activations are kept in INT4. This by itself is not novel and susceptible to excess noise. The proposed method is to enhance this quantization using learned transformations and other techniques.

**Strengths:**

-CEAVQ provides formulas for updating weights to account for activation quantization.

**Weaknesses:**

- The proposed learned transformation in Section 3.1 is given by T=O\Lambda, i.e, a product of a rotation matrix and a smoothing matrix. An expression for the MSE of the layer's gemm output is given in terms of the weight and activation statistics. Then it is claimed that the transformation is theoretically guaranteed to minimize this MSE with a promised proof in the appendix. But the actual construction of T is not given. Therefore, the claim is essentially void.
- The paper assumes INT4 tensor cores. These have been discontinued since the Ampere architecture. In Blackwell, there are NVFP4 tensor cores which may be more fit to explore.

**Questions:**

- Please provide a construction for the learned transformation T.
- Shouldn't CEAVQ also account for weight quantization noise?
- Please explain why the evaluations of related works are not similar to those in the corresponding papers, e.g., quarot is claimed to have a perplexity in the 5e4 regime - but that paper itself claims good accuracy for extremely low bitwidth quantization.

---

> ### Author Response · Authors · 2025-11-21
> **Rebuttal for Reviewer HrU8. Part [1]**
>
> **Dear Reviewer HrU8,**
>
> We sincerely thank you for your thorough evaluation and constructive comments. We address your valuable comments point by point as follows:
>
> ---
>
>
>
> > The proposed learned transformation in Section 3.1 is given by T=O\Lambda, i.e, a product of a rotation matrix and a smoothing matrix. An expression for the MSE of the layer's gemm output is given in terms of the weight and activation statistics. Then it is claimed that the transformation is theoretically guaranteed to minimize this MSE with a promised proof in the appendix. But the actual construction of T is not given. Therefore, the claim is essentially void.
>
> We thank the reviewer for the careful reading of Section 3.1 and for pointing out the lack of clarity regarding the theoretical guarantee and the construction of the transformation $T$. This comment indeed reveals an imprecision in our current wording. Below we clarify **(1) what is actually proved**, and **(2) how the theoretical construction relates to the practical implementation**.
>
>
> **(1) What do we actually prove?**
>
> **In Proposition 1 of Appendix A.2**, our analysis focuses on the unified error proxy defined in Equations (4)/(10) **rather than the global minimum of the original MSE**. More precisely (as detailed in A.2.5):
> - **Assumptions**: Let the input covariance be $H = \mathbb{E}[xx^\top] \succ 0$.
> - **Constructio**n: We construct a transformation $T^\star = H^{-1/2} O^\star$, where $O^\star$ is an energy-equalizing orthogonal matrix (based on Lemma 2).
> - **Conclusion**: Under these assumptions, we prove that $\Delta E = E(T^\star) - E(I) < 0$. This indicates that, compared to the identity case $T = I$, the transformation $T^\star$ simultaneously reduces the unified proxy comprising activation SQ error, weight VQ error, and codebook integer quantization error.
>
> Therefore, our theoretical conclusion establishes **the existence of a class of transformations $T^\star = \Lambda^\star O^\star$ that strictly reduces the unified error in Equation (10)**. We do not claim to have derived the explicit global closed-form solution that minimizes this error. In the revised version, we will emphasize "provably reduce the unified error under the stated assumptions" to avoid the misinterpretation of "global minimization".
>
> **(2) Theoretical Construction and Practical Implementation of $T$**
>
>
> Appendix A.2 provides more than just an existence theorem. It identifies a specific candidate construction that satisfies the conditions:$$T^\star = H^{-1/2} O^\star, \quad O^\star \in \mathcal{O}(d)$$
>
> - The term $H^{-1/2}$ corresponds to the whitening transformation $\Lambda ^\star$ (Lemma 1), which balances the output covariance and thereby minimizes the VQ-related proxy.
> - The term $O^\star$ corresponds to the energy-equalizing rotation (Lemma 2), which reduces the $\ell_\infty$ norm and consequently compresses the dynamic range required for scalar quantization and codebook quantization.
>
>
> So Appendix A.2 is not merely a pure existence result, it explicitly singles out a concrete form $𝑇^\star$ with provable error-reduction properties for the unified proxy.
>
>
> In terms of **practical implementation**, we do not explicitly compute the exact Hessian inverse $H^{-1/2}$ or the analytical rotation $O^\star$. Instead, we **adopt a learnable parameterization $T = \Lambda Q$ and optimize it on limited calibration data to approximate the theoretical objective described above**. As briefly mentioned in Section 4 (Baselines and Implementation Details):
> - We parameterize $T$ using a diagonal matrix $\Lambda$ and an orthogonal matrix $Q$, maintaining the approximate orthogonality of $Q$ during training.
> - Following the optimization approach in OSTQuant (Hu et al., 2025), we iteratively update $T$ via an end-to-end distribution matching objective. This minimizes the discrepancy between the output distribution of the mixed SQ–VQ pipeline and that of the full-precision model.This optimization objective is highly correlated to the unified MSE proxy discussed in Appendix A.2. Consequently, the learned $T$ empirically converges to a transformation that approximates the properties of "whitening + energy-equalizing" (as evidenced by the empirical results in Figure 2(b,c)).
>
> To address these points in the revised manuscript, we will:
> - Explicitly state the construction $T^\star = H^{-1/2} O^\star$ in Section 3.1 to address the reviewer's concern about the "void" claim.
> - Add a summary sentence at the beginning of Appendix A.2 clarifying that this is a concrete construction, not merely an existence proof.

---

> > ### Author Response · Authors · 2025-11-21
> > **Rebuttal for Reviewer HrU8. Part [2]**
> >
> > > The paper assumes INT4 tensor cores. These have been discontinued since the Ampere architecture. In Blackwell, there are NVFP4 tensor cores which may be more fit to explore.
> >
> > We thank the reviewer for raising this very practical point regarding hardware evolution and for highlighting the relevance of NVFP4 tensor cores on Blackwell-class GPUs. We fully agree that NVFP4 and related emerging 4-bit formats are highly promising targets for future deployment. Below we clarify (a) **why we chose INT4 as the working setting in the paper**, (b) **why AEC-SVQ is essentially format-agnostic across 4-bit representations**, and (c) **how we validate this claim with additional NVFP4-based experiments**.
> >
> > **（1）Rationale for Choosing INT4**
> >
> > Our choice to focus on INT4 was driven by two primary factors:
> > - **Standardization and Fairness**: Our primary motivation was to position this work within the widely adopted general-purpose 4-bit quantization landscape. Consequently, we selected INT4 as it remains the standard setting in current academic literature. **This choice ensures that our comparisons with existing SOTA methods are both direct and fair**.
> > - **Broad Hardware Applicability**: While specific high-end Data Center GPUs are moving toward FP4, **4-bit integer arithmetic remains the standard for a vast array of edge devices, NPUs, and mobile accelerators.** In our paper, the term "INT4 Tensor Cores" serves as a proxy for generic 4-bit integer matrix multiplication units, which are widely available across the hardware landscape.
> >
> > **(2) Format-agnostic nature of AEC-SVQ**
> >
> > - **Core Philosophy**: The fundamental principle of AEC-SVQ is to map continuous distributions onto a finite 4-bit discrete set via unified learnable transformations and SQ/VQ designs.
> > - **Mathematical Compatibility**: This framework is mathematically transferable to 4-bit floating-point representations such as NVFP4. The migration requires only two minor adjustments: replacing the uniform quantization grid of the activation SQ with the non-uniform NVFP4 scale, and aligning the weight codebook target points with the NVFP4 grid.
> > - **Unified Mechanism**: Crucially, the unified error proxy, the CEAVQ objective, and the Adaptive Compensation mechanism remain unchanged. Therefore, we consider AEC-SVQ to be inherently format independent. Its effectiveness relies on the available 4-bit dynamic range and encoding structure rather than the specific underlying data type (INT4 versus NVFP4)
> >
> > **(3) Empirical Verification on NVFP4**
> >
> > To substantiate this claim, we conducted an experiment simulating NVFP4 backend. As shown in **Table R3-1**, AEC-SVQ maintains its superior performance on NVFP4, demonstrating that the method is effective regardless of the underlying 4-bit representation.
> >
> >
> > **Table R3-1: Performance Comparison of AEC-SVQ under Simulated NVFP4 and INT4 Formats.**
> > | model     | Formats        | Bits | Wiki-PPL (↓) | ARC-c (↑) | ARC-e (↑) | BoolQ (↑) | HellaS. (↑) | OBQA (↑) |  PIQA (↑) | SIQA (↑) | WinoG. (↑) | Avg. (↑) |
> > |:---------:|:--------------:|:----:|:------------:|:---------:|:---------:|:---------:|:-----------:|:--------:|:---------:|:--------:|:----------:|:--------:|
> > | llama3-8b | Full Precision | W2A4 | 6.14         | 53.50     | 77.74     | 81.10     | 79.18       | 44.80    | 80.63     | 47.08    | 73.01      | 67.13    |
> > |           | INT4           | W2A4 | 8.65         | 41.30     | 63.64     | 74.59     | 69.04       | 38.40    | 74.21     | 42.37    | 63.61      | 58.39    |
> > |           | NVFP4          | W2A4 | 8.26         | 42.88     | 64.24     | 75.88     | 68.55       | 37.60    | 75.90     | 42.75    | 65.01      | 59.10    |

---

> > > ### Comment · Reviewer_HrU8 · 2025-11-25
> > > **Responses re. 4 bit formats**
> > >
> > > (1) rationale: please provide examples of devices running LLMs today that have the INT4 tensor cores you are mentioning.
> > >
> > > (2) format agnostic method: with NVFP4 it's not just a set of 4-bit discrete values. The block scale factors introduce a non-trivial difference. Does the work that into account?
> > >
> > > (3) new empirical results: sounds good thanks.

---

> > > > ### Comment · Reviewer_HrU8 · 2025-11-25
> > > > **Maintain the score for now**
> > > >
> > > > I am thankful for the various responses. However, I did not find the responses to the weaknesses convincing enough. For now I maintain the score of 4.

---

> > ### Comment · Reviewer_HrU8 · 2025-11-25
> > **Thanks for the clarification**
> >
> > I appreciate the concession. Let's be sure to update the manuscript with this information in order to not deceive the reader. It is not surprising at all that a rotation would strictly improve upon the quantization noise compared the vanilla transformation-less quantization.

---

> ### Author Response · Authors · 2025-11-21
> **Rebuttal for Reviewer HrU8. Part [3]**
>
> > Shouldn't CEAVQ also account for weight quantization noise?
>
> We appreciate the reviewer for raising this critical question regarding whether CEAVQ explicitly accounts for weight quantization noise. This comment demonstrates a thorough engagement with our derivation. We wish to clarify that the design of **CEAVQ indeed incorporates weight quantization noise**, as it **explicitly models both weight and activation quantization noise simultaneously within its foundational formulation**.
>
> We elucidate this mechanism from two complementary perspectives: **theoretical formulation** and **algorithmic implementation**.
>
> **(1) Unified proxy includes both weight and activation quantization errors.**
>
>
> As derived in Appendix A.3, we define the proxy loss function (Eq. 12) as:
> $$\ell(\hat{W}) = \mathbb{E}_X \left\| (\hat{W} - W)X - W(\tilde{X} - X) \right\|_F^2$$
>
> This objective explicitly comprises two distinct components:
>
> - **Weight Error Term:** $(\hat W - W)X$ represents the error propagated to the layer output due to **current layer's weight quantization**. This includes deviations caused by both VQ and codebook integer quantization.
> - **Activation Error Term:** **$W(\tilde X - X)$ represents the error propagated to the layer output due to activation quantization.** Here, $(\tilde X - X)$ captures the cumulative perturbations caused by **upstream weight and activation quantization**.
>
> Therefore, this proxy is not limited to activation noise. Instead, it explicitly couples the **weight quantization error of the current layer** with the **cumulative upstream weight and activation quantization errors** within a single objective function.
>
> Solving this proxy $w.r.t.\hat W$ yields the continuous optimum (after Eq. (19)):
> $$W_{\text{opt}} = W + W G H^{-1}$$
> where
> $$H = \mathbb{E}[XX^\top], \quad G = \mathbb{E}[(\tilde X - X)X^\top]$$
> Here $G$ depends explicitly on the statistics of $\tilde X - X$(activation-side accumulated error). **The weight quantization noise is already explicitly represented by $(\hat W - W)$ in the proxy and is controlled through optimizing $\hat W$ itself.** Since $\hat W$ is the decision variable, its deviation from $W$ is inherently modeled and minimized through the objective function.
>
> **(2) Implementation Details: Dual Mitigation of Weight and Activation Noise**
>
> In the practical column-wise quantization algorithm (Eq.(6) in the paper), the update rule is formulated as follows:$$\hat W_k = Q\Big( W_k \;+\; (W_{1:(k-1)} - \hat W_{1:(k-1)}) a_k \;+\; (W G H^{-1})_k \Big)$$This formulation consists of three distinct components:
>
> - **Original Weight**: $W_k$ is the target weight for the current column.
> - **Weight Quantization Noise Correction:** The term $(W_{1:(k-1)} - \hat W_{1:(k-1)}) a_k$ is the standard residual feedback. It explicitly **corrects the weight quantization errors accumulated from preceding columns**.
> - **Cumulative Error Compensation:** The term $(W G H^{-1})_k$ is our cumulative error compensation term. It captures the comprehensive **impact of upstream weight and activation quantization** on the current layer input distribution via $\tilde X - X$.
>
> Thus, CEAVQ simultaneously leverages:
> - **Weight noise correction** via residual feedback $(W_{1:(k-1)} - \hat W_{1:(k-1)}) a_k$
> - **Activation/upstream accumulated noise compensation** through the closed-form $W G H^{-1}$.
>
> > Please explain why the evaluations of related works are not similar to those in the corresponding papers, e.g., quarot is claimed to have a perplexity in the 5e4 regime - but that paper itself claims good accuracy for extremely low bitwidth quantization.
>
> We appreciate the reviewer’s careful attention to the baseline comparisons. We wish to clarify that this discrepancy is not due to implementation errors, but rather stems from the significantly more **aggressive W2A4 quantization regime** targeted in our work, which **exceeds the operating range of methods designed for W4A4**.
>
> Our main evaluation (**Table 1**) strictly adheres to a **W2A4 regime**. Methods like QuaRot demonstrate excellent accuracy in their original settings (typically W4A4，W4A8 or weight-only), **they were not originally designed to handle the severe loss of 2-bit weights combined with 4-bit activations.** When we rigorously adapted these methods to this harsher W2A4 setting, we observed that models suffer from rapid degradation, leading to the perplexity in the $10^4$ range reported in our table.
>
> We explicitly confirm that these results are **fully reproducible** using the official implementations of the baseline methods. We **did not** modify the internal logic of the baselines but simply adapted the configuration parameters to match our target W2A4 setting.
>
> For instance, to reproduce the QuaRot results reported in our paper, we utilized the official repository (https://github.com/spcl/QuaRot) with the following arguments:
> ```
> python main.py --model meta-llama/Llama-2-7b-hf --rotate --a_bits 4 --v_bits 4 --k_bits 4 --w_bits 2
> ```

---

> ### Author Response · Authors · 2025-11-27
> **Clarifications on Novelty and NVFP4 Relevance. Part[1]**
>
> **Dear Reviewer HrU8,**
>
> We sincerely thank you for your timely follow-up and constructive engagement. **We are pleased to note your acknowledgment of our clarifications in the previous round.**
>
> In this response, we: (1) **Clarify our theoretical claims** to address specific concerns. (2) **Provide further clarification on the core contributions of our work** to explicitly distinguish it from prior art. (3) **Continue the discussion regarding the NVFP4 format** and hardware relevance, demonstrating the compatibility of our approach.
>
> We remain fully committed to addressing your concerns to the best of our ability.
>
> ### **1 Clarification on Theoretical Claims**
>
> We are glad to have reached a consensus on this point. We wish to respectfully clarify that our **original submission never claimed** that the transformation theoretically guarantees the minimization of MSE. Rather, **our theoretical conclusion establishes the existence of a class of transformations $T^\star = \Lambda^\star O^\star$ that strictly reduces the unified error defined in Equation (10).**
>
> As promised, and to ensure absolute clarity, we will revise the final manuscript to **emphasize that our method "provably reduces the unified error under the stated assumptions" and will explicitly state the construction $T^\star = H^{-1/2} O^\star$.**
>
> ### **2 Further Clarification on Core Contributions**
>
> Regarding your comment that "it is not surprising at all that a rotation would strictly improve quantization", we seek to clarify that **rotation alone is insufficient for the extreme settings we target**.
>
> While we acknowledge that rotation helps smooth outliers, our empirical results (**Table 1**) demonstrate that **relying solely on rotation fails in the W2A4 regime**. SOTA rotation-based methods (e.g., QuaRot, SpinQuant, OSTQuant) suffer significant degradation under these constraints. **This proves that without systematically modeling the hybrid W2A4 framework, simple rotation is inadequate**.
>
> **AEC-SVQ achieves SOTA performance** not because we apply a "simple rotation," but because we introduce **three targeted innovations:**
>
>
>  $\star$ **Innovation I: Unified Optimization for Hybrid SQ-VQ with Learned Transformation**
>
> - **Limitations of existing approaches**
>
>   These two lines are largely developed in isolation:
>   - SQ-based methods transform and scale representations mainly to benefit scalar weight/activation quantization, **without addressing the structure needed for VQ**.
>   - VQ-based methods optimize weight compression, typically with **high-precision activations and floating-point or complex decode operators**, and do not account for activation quantization or integer kernel structure.
>   - As a result, in an extreme W2A4 setting, **it is difficult for existing methods to simultaneously achieve 2-bit weight storage and truly deployable 4-bit integer matmul**.
> - **Our Novel Contribution**
>
>   Our contribution is to turn the learned transform into a **single, unified mechanism** that simultaneously serves **Weight VQ，Activation SQ and INT4 codebook alignment**.
>   - We introduce a learnable linear transform $T$ that is explicitly designed as a joint shaping transform:
>     - In the $T$-domain, **weights become more isotropic and better conditioned for VQ**.
>     - In the $T^{-1}$-domain, activations obtain a contracted dynamic range that is more amenable to **high-fidelity scalar quantization**.
>   - We analyse a unified proxy objective (Eqs. (5)/(12) in the paper) under W2A4 inference constraints, and show how a single learnable $T$ can **simultaneously tighten upper bounds on weight VQ error, activation SQ error, and integerization error**.
>   - When designing the codebook, we explicitly incorporate INT4 GEMM structure so that the **quantized weights map directly to INT4 tensor-core–friendly kernels**, rather than treating VQ as a standalone compression module.
>
> Thus, our contribution lies in the **unified theoretical and algorithmic framework that co-optimizes weight VQ, activation SQ, and INT4 codebooks** under a single transform and objective, enabling practical W2A4 integer inference for large LLMs.

---

> > ### Author Response · Authors · 2025-11-27
> > **Clarifications on Novelty and NVFP4 Relevance. Part[2]**
> >
> > $\star$ **Innovation II: CEAVQ (Cumulative-Error-Aware Vector Quantization)**
> >
> >
> > - **Limitations of existing methods**
> >
> >   Under stringent W2A4 constraints, the error contribution is not limited to the weight quantization of the current layer but stems from two additional sources:
> >   - The **cumulative quantization error** propagated from all preceding upstream layers.
> >   - The **distributional shift caused by compressing activations to low-precision** 4-bit representations.
> >
> >   Conventional GPTQ-style local proxies typically restrict optimization to the discrepancy between the current layer's weights $W$ and their quantized counterparts $\hat{W}$. Consequently, these methods **fail to explicitly model how accumulated errors propagate through deep networks to influence the optimal form of compensation.** Furthermore, they overlook the critical coupling between these quantization errors and the design of the VQ codebook.
> > - **Our Novel Contribution（CEAVQ）**
> >
> >   Our CEAVQ formulation advances this line through **two primary contributions**:
> >   - **Cumulative-Error-Aware Objective**: We propose a proxy objective that explicitly incorporates **both upstream propagation errors and activation quantization errors.** In **Appendix A.3**, we derive **the closed-form optimal solution** for this objective as $W_{\text{opt}} = W + W G H^{-1}$.  Consequently, **it transcends the conventional setting** which is restricted to the current layer and assumes a single quantization error source.
> >   - **Integration with Vector Quantization**: We implement CEAVQ by leveraging this theoretical foundation. We first calculate the **cumulative-error-aware target $W_{\text{opt}}$ in the continuous domain** and subsequently project it onto the VQ codebook using a **column-wise, feedback-aware VQ procedure.** This step effectively integrates cumulative error compensation directly into the VQ optimization process.
> >
> >   To the best of our knowledge, this strategy of **deriving a closed-form solution for cumulative errors and integrating it with vector quantization has not been explored in prior LLM quantization work** and transcends the limitations of GPTQ-style local compensation.
> >
> > $\star$ **Innovation III: Adaptive bias–variance shrinkage for limited calibration data**
> >
> > - **Limitation we address.**
> >   - **Susceptibility to Overfitting**: Under conditions of limited calibration data, the estimation of matrices $G$ and $H$ inevitably contains noise. Consequently, directly applying the theoretically unregularized compensation term $W G H^{-1}$ tends to overfit the calibration set. This leads to **significant performance instability** when the model is evaluated on **real data distribution**.
> >   - **Reliance on Heuristics**: Common approaches in existing studies typically **rely on fixed scaling coefficients or empirical hyperparameter tuning.** They fail to derive a column-wise, adaptive closed-form shrinkage solution from a consistent theoretical proxy.
> > - **Our Novel Contribution**
> >   - We formulate the problem as a **bias–variance trade-off** on top of the cumulative-error-aware proxy and derive a **closed-form, column-wise shrinkage solution**:$\hat W_k = W_k + \alpha_k (W G H^{-1})_k$, where $\alpha_k$ is an adaptive shrinkage factor tied to the estimated noise variance of column $k$.
> >   - Propose a nearly hyperparameter-free rule for $\alpha_k$ as a function of the estimated variance(see Appendix A.4 for details), so that **high-noise columns receive stronger shrinkage (more conservative correction), and low-noise columns retain stronger compensation**.
> >
> >   Conceptually, we **derive a column-wise optimal shrinkage solution directly from the unified cumulative-error proxy and regularized objective**. This solution is subsequently employed to stabilize CEAVQ under realistic calibration budgets. To the best of our knowledge, this represents **the first introduction of a variance-aware, column-wise shrinkage mechanism specifically tailored for cumulative-error compensation in LLM quantization**.

---

> > > ### Author Response · Authors · 2025-11-27
> > > **Clarifications on Novelty and NVFP4 Relevance. Part[3]**
> > >
> > > ### **3 Clarifications on Hardware and NVFP4 Details**
> > > We specifically address your latest comments regarding: **(1) concrete examples of hardware currently running LLMs with INT4 Tensor/Matrix capabilities**, and **(2) the relationship between NVFP4 block scales and our "format-agnostic" claim**.
> > >
> > > **(1) Hardware Instances Supporting INT4**
> > >
> > > we would like to clarify that while high-end data center GPUs (e.g., Hopper/Blackwell) are transitioning toward FP8/FP4/NVFP4, **INT4 remains a key design point for achieving energy efficiency and low memory footprint on consumer GPUs and mobile/edge devices.** For these platforms, **memory capacity is often a more critical bottleneck than latency.** This is exactly the regime where W2A4 quantization is most relevant.
> > >
> > > Specific hardware examples include:
> > > - **Consumer GPUs (NVIDIA RTX 40 Series / Ada Lovelace)**: The Ada Lovelace architecture (e.g., RTX 4090) natively supports INT4 Tensor Cores. In practice, **these are among the most widely used GPUs for local LLM inference**. The open-source community extensively relies on 4-bit integer quantization to fit large models (e.g., LLaMA-3-70B, DeepSeek) into consumer VRAM.
> > > - **Mobile NPUs (On-Device AI)**: This is the primary domain for low-bit quantization. Flagship SoCs such as the **Qualcomm Snapdragon 8 Gen 3** and **MediaTek Dimensity 9300** feature NPUs with **native INT4 acceleration**. **For battery-constrained devices, INT4 offers a superior TOPS/W (operations per watt) ratio compared to floating-point formats**.
> > > - **Edge & Embedded Systems**: The **NVIDIA Jetson Orin family** integrates **Ampere-based GPUs where Tensor Cores support INT4 integer arithmetic**. NVIDIA’s own deployment examples utilize INT4 to run Vision-Language Models (VLMs) and LLMs on these edge boards.
> > > - **Data Center GPUs (Current & Legacy)**: While new architectures are emerging, widely deployed GPUs like NVIDIA T4, A100, L4, and L40S explicitly list INT4 Tensor Core (or INT4 Matrix) peak performance. They remain the workhorses for many production-grade 4-bit LLM deployments.
> > >
> > > Based on these examples, we emphasize the following:
> > > - **Relevance of INT4**: While the latest generation of architectures (e.g., Blackwell) prominently features new formats such as NVFP4 and FP4, **the hardware ecosystem supporting INT4 tensor/matrix operations remains ubiquitous today**, particularly within the **Edge and NPU domains** where practical LLM inference is widely deployed.
> > > - **Rationale for Experimental Setting**: Our decision to adopt "INT4" as the primary experimental setting was driven by **the need to align with prevailing 4-bit integer quantization research** and real-world deployment standards, **rather than a reliance on obsolete or niche hardware components.**
> > > - **Generalizability to New Formats**: As articulated in our rebuttal, our algorithmic framework subject to the constraints of a given 4-bit quantization operator $Q$ (whether INT4 or NVFP4). **Our experimental validation on NVFP4 confirms that the proposed method is format-agnostic** and readily adaptable to emerging 4-bit floating-point standards.

---

> > > > ### Author Response · Authors · 2025-11-27
> > > > **Clarifications on Novelty and NVFP4 Relevance. Part[4]**
> > > >
> > > > **(2) Compatibility with NVFP4 Block Scales**
> > > >
> > > > We appreciate the reviewer highlighting the importance of block scale factors in the NVFP4 format. We confirm that our method is **fully compatible with NVFP4, including its block scale factors**.
> > > >
> > > >
> > > > * **Structural Similarity between NVFP4 and INT4:** Mathematically, both NVFP4 and standard INT4 quantizers share a common structural definition: a linear scaling operation followed by a discrete mapping.
> > > >     - **In NVFP4**: The block scale functions as a local linear transformation applied to a specific weight block. This operation rescales the magnitude without altering the fundamental distributional shape or introducing non-linear shifts prior to mapping values onto the 4-bit floating-point grid.
> > > >     - **In INT4**: Standard implementations similarly utilize per-channel or per-group scaling. Therefore, conceptually, the **NVFP4 block scale is analogous to a fine-grained group scale in INT4. Both can be modeled within a unified framework.**
> > > > * **Sensitivity of Block Scaling to Outliers:** We contend that **block-wise scaling is inherently susceptible to outliers.** A single outlier necessitates a larger scaling factor, thereby collapsing the effective resolution for the remaining elements within the constrained 4-bit grid. While FP4 offers a wider dynamic range than INT4, it does not fundamentally alleviate this **outlier dominance**. This is precisely the issue **AEC-SVQ aims to mitigate at the distributional level**.
> > > > * **Full Compatibility of the AEC-SVQ Framework with NVFP4:** We clarify that our method adapts to NVFP4 not only through distributional shaping but also via the fundamental universality of our optimization framework.
> > > >     * **End-to-End Error Awareness**: Since $T$ is optimized, it implicitly learns the high error cost associated with outlier-dominated blocks. Consequently, the optimization process drives $T$ to rotate the feature space, effectively redistributing outliers across channels to smooth local blocks.
> > > >     * **Learned $T$ for Format Fidelity**: By reshaping the input distribution to be more compact and regular via $T$, we not only facilitate **global uniform quantization** (e.g., INT4) but also **maximize the fidelity of format-specific quantization, including NVFP4 with block scales**. In essence, when the input distribution is rendered "quantization-friendly," the block scales of **NVFP4 can more effectively capture local signal variations without being distorted by outliers**.
> > > >     * **Format Agnosticism of the Framework**: Crucially, **the Unified Error Proxy, the CEAVQ objective, and the Adaptive Compensation mechanism remain structurally unchanged**. These components are jointly optimized with the actual quantization operator. As long as the quantization process can be formulated as a composition of **linear scaling and discrete mapping**, the block scale factors are absorbed into the quantization operator. Thus, **NVFP4 block scales does not invalidate our theoretical analysis, it merely alters the specific realization of the quantization operator.**
> > > >
> > > >
> > > > ### **Conclusion**
> > > > We hope these clarifications regarding the **theoretical guarantees**, the three distinct **technical innovations beyond simple rotation**, and the **compatibility with NVFP4** fully address your concerns. **Given these clarifications, we respectfully hope you might reconsider the evaluation of our work.**

---

> ### Author Response · Authors · 2025-11-28
> **Gentle Follow-up in the Final Week of Discussion**
>
> **Dear Reviewer HrU8,**
>
> Thank you again for your time and for the constructive, insightful comments you provided on our submission. As the discussion period is drawing to a close, we are writing to kindly follow up to see if you have had the opportunity to review our previous response.
>
> In our detailed rebuttal, we specifically addressed your concerns by:
> -  **Clarifying our theoretical claims:** We confirmed that our theoretical conclusion establishes that the transformation $T^\star$ strictly reduces the **unified error** (Eq. 10), rather than guaranteeing global MSE minimization. We have committed to explicitly stating the construction $T^\star = H^{-1/2} O^\star$ in the final revision.
> -  **Distinguishing our contribution from simple rotation:** We elaborated on why rotation alone fails in the W2A4 regime and detailed our three specific innovations—**(I) Unified Optimization, (II) CEAVQ (Cumulative-Error-Aware VQ), and (III) Adaptive Bias-Variance Shrinkage**—which collectively solve the hybrid error modeling problem that rotation-based methods cannot address.
> -  **Addressing Hardware and NVFP4 relevance:** We provided concrete examples of current hardware reliant on INT4 (e.g., Consumer GPUs, Mobile NPUs) and demonstrated mathematically that our framework is **format-agnostic and fully compatible with NVFP4 block scales**, as the optimization inherently handles block-wise outlier distributions.
>
> We completely understand that you may have many commitments, and we are truly grateful for any time you could spare. We remain fully available to answer any further questions you may have. We respectfully hope you might reconsider the evaluation of our work.
>
> Thank you again for your thoughtful evaluation and valuable insights.
>
> Best regards,
>
> The Authors

---

### Official Review · Reviewer_vGbd · 2025-10-31

**Soundness:** 4
**Presentation:** 3
**Contribution:** 2
**Rating:** 6
**Confidence:** 4

**Summary:**

This paper proposes a method that combines scalar quantization (SQ) and vector quantization (VQ) to quantize large language models (LLMs) to W2A4 precision. The approach employs a learned rotation-smooth transformation and a cumulative error aware vector quantization technique, supported by a layer-wise correction factor. Through this design, AEC-SVQ achieves both high compression rates and computational efficiency.

**Strengths:**

This LLM quantization paper presents a hybrid approach that combines SQ and VQ with a learned transformation, leading to an intuitive understanding of its benefits, supported by clear mathematical derivations. It also provides a detailed study that examines various aspects and techniques across multiple models. Additionally, it adopts a layer-wise regularization factor to prevent overfitting in the compensation process.

**Weaknesses:**

Please refer to the questions below.

**Questions:**

1. Beyond the combination of existing SQ and VQ techniques, what are the main novelties of this paper? The use of a learned rotation matrix has already been explored in prior works such as SpinQuant and OSTQuant, and the CEA column-wise compensation based on WX error appears similar to GPTQ, not to mention the use of codebook quantization. Please clarify which components are borrowed or inspired by prior work and which constitute the novel contributions of this paper.

2. How sensitive is your method to the choice of the calibration dataset?

3. Fine-tuning appears to have a significant impact on the results, which raises some concerns about the robustness of the proposed strategy. Could you elaborate on this point and confirm whether the comparisons with other works are conducted fairly (e.g., fine-tuning applied to your method but not to baselines)?

4. In Section A.5.2, is the analysis referring to OSTQ?

---

> ### Author Response · Authors · 2025-11-21
> **Rebuttal for Reviewer vGbd. Part [1]**
>
> **Dear Reviewer vGbd,**
>
> We sincerely thank you for your thorough evaluation and constructive comments. We address your valuable comments point by point as follows:
>
> > Beyond the combination of existing SQ and VQ techniques, what are the main novelties of this paper? The use of a learned rotation matrix has already been explored in prior works such as SpinQuant and OSTQuant, and the CEA column-wise compensation based on WX error appears similar to GPTQ, not to mention the use of codebook quantization. Please clarify which components are borrowed or inspired by prior work and which constitute the novel contributions of this paper.
>
>
> We thank the reviewer for the careful assessment of our paper’s novelty. We clarify the relationship between our work and prior art, the limitations of existing methods, and our three specific innovations:
> - **Unified Optimization for Hybrid SQ-VQ with Learned Transformation**
> - **CEAVQ (Cumulative-Error-Aware VQ)**
> - **Adaptive bias–variance shrinkage for limited calibration data**
>
> We further demonstrate the effectiveness of these contributions through extensive experiments in both the W2A4 and 2-bit weight-only settings.
>
> ### **Innovation I: Unified Optimization for Hybrid SQ-VQ with Learned Transformation**
> - **What is inspired by prior work**：
>   - **SpinQuant, OSTQuant, QuaRot** and related methods have shown that **applying a learned rotation / linear transform to intermediate representations can mitigate outliers** and improve scalar quantization of weights/activations.
>   - **QuIP#, AQLM, PCDVQ** and others have demonstrated the effectiveness of vector quantization with codebooks for compressing weights, usually in **weight-only** settings and largely **decoupled from activation quantization** and from the structure of integer GEMM kernels.
> - **Limitations of existing approaches**
>
>     These two lines are largely developed in isolation:
>   - SQ-based methods transform and scale representations mainly to benefit scalar weight/activation quantization, **without addressing the structure needed for VQ**.
>   - VQ-based methods optimize weight compression, typically with **high-precision activations and floating-point or complex decode operators**, and do not account for activation quantization or integer kernel structure.
>   - As a result, in an extreme W2A4 setting, **it is difficult for existing methods to simultaneously achieve 2-bit weight storage and truly deployable 4-bit integer matmul**.
> - **Our Novel Contribution**
>
>   Our contribution is to turn the learned transform into a **single, unified mechanism** that simultaneously serves **Weight VQ，Activation SQ and INT4 codebook alignment**.
>   - We introduce a learnable linear transform $T$ that is explicitly designed as a joint shaping transform:
>     - In the $T$-domain, **weights become more isotropic and better conditioned for VQ**.
>     - In the $T^{-1}$-domain, activations obtain a contracted dynamic range that is more amenable to **high-fidelity scalar quantization**.
>   - We analyse a unified proxy objective (Eqs. (5)/(12) in the paper) under W2A4 inference constraints, and show how a single learnable $T$ can **simultaneously tighten upper bounds on weight VQ error, activation SQ error, and integerization error**.
>   - When designing the codebook, we explicitly incorporate INT4 GEMM structure so that the **quantized weights map directly to INT4 tensor-core–friendly kernels**, rather than treating VQ as a standalone compression module.
>
> Thus, while the idea of a learned rotation and the existence of codebook quantization are not new, our contribution lies in the **unified theoretical and algorithmic framework that co-optimizes weight VQ, activation SQ, and INT4 codebooks** under a single transform and objective, enabling practical W2A4 integer inference for large LLMs.

---

> ### Author Response · Authors · 2025-11-21
> **Rebuttal for Reviewer vGbd. Part [2]**
>
> ### **Innovation II: CEAVQ (Cumulative-Error-Aware Vector Quantization)**
>
> - **What is inspired by GPTQ-style methods.**
>   GPTQ optimizes a local objective of the form $\|(W - \hat W)X\|_F^2$, using a Hessian approximation and performing column-wise compensation.
> - **Limitations of existing methods**
>   Under stringent W2A4 constraints, the error contribution is not limited to the weight quantization of the current layer but stems from two additional sources:
>   - The **cumulative quantization error** propagated from all preceding upstream layers.
>   - The **distributional shift caused by compressing activations to low-precision** 4-bit representations.
>
>   Conventional GPTQ-style local proxies typically restrict optimization to the discrepancy between the current layer's weights $W$ and their quantized counterparts $\hat{W}$. Consequently, these methods **fail to explicitly model how accumulated errors propagate through deep networks to influence the optimal form of compensation.** Furthermore, they overlook the critical coupling between these quantization errors and the design of the VQ codebook.
> - **Our Novel Contribution（CEAVQ）**
>
>    Our CEAVQ formulation advances this line through two primary contributions:
>   - **Cumulative-Error-Aware Objective**: We propose a proxy objective that explicitly incorporates both upstream propagation errors and activation quantization errors. In Appendix A.3, we derive the closed-form optimal solution for this objective as $W_{\text{opt}} = W + W G H^{-1}$,where $H = \mathbb{E}[XX^\top]$ and $G = \mathbb{E}[(\tilde X - X)X^\top]$. Here, $X$ denotes the representation already perturbed by upstream quantization and activation quantization, while $\tilde X$ represents the full-precision floating-point representation.  This formulation mathematically characterizes how cumulative errors alter the optimal pre-compensated weights at the current layer. Consequently, it transcends the conventional GPTQ setting which is restricted to the current layer and assumes a single quantization error source.
>   - **Integration with Vector Quantization**: We implement CEAVQ by leveraging this theoretical foundation. We first calculate the cumulative-error-aware target $W_{\text{opt}}$ in the continuous domain and subsequently project it onto the VQ codebook using a column-wise, feedback-aware VQ procedure. This step effectively integrates cumulative error compensation directly into the VQ optimization process.
>
>   To the best of our knowledge, this strategy of **deriving a closed-form solution for cumulative errors and integrating it with vector quantization has not been explored in prior LLM quantization work** and transcends the limitations of GPTQ-style local compensation.
>
> ### **Innovation III: Adaptive bias–variance shrinkage for limited calibration data**
>
> - **Relationship to Prior Work**: **None**. To the best of our knowledge, no adaptive compensation algorithms specifically designed for scenarios with limited calibration data have been proposed before.
> - **Limitation we address.**
>   - **Susceptibility to Overfitting**: Under conditions of limited calibration data, the estimation of matrices $G$ and $H$ inevitably contains noise. Consequently, directly applying the theoretically unregularized compensation term $W G H^{-1}$ tends to overfit the calibration set. This leads to **significant performance instability** when the model is evaluated on **real data distribution**.
>   - **Reliance on Heuristics**: Common approaches in existing studies typically **rely on fixed scaling coefficients or empirical hyperparameter tuning.** They fail to derive a column-wise, adaptive closed-form shrinkage solution from a consistent theoretical proxy.
> - **Our Novel Contribution**
>   - We formulate the problem as a **bias–variance trade-off** on top of the cumulative-error-aware proxy and derive a **closed-form, column-wise shrinkage solution**:$\hat W_k = W_k + \alpha_k (W G H^{-1})_k$, where $\alpha_k$ is an adaptive shrinkage factor tied to the estimated noise variance of column $k$.
>   - Propose a nearly hyperparameter-free rule for $\alpha_k$ as a function of the estimated variance(see Appendix A.4 for details), so that **high-noise columns receive stronger shrinkage (more conservative correction), and low-noise columns retain stronger compensation**.
>
>   Conceptually, we **derive a column-wise optimal shrinkage solution directly from the unified cumulative-error proxy and regularized objective**. This solution is subsequently employed to stabilize CEAVQ under realistic calibration budgets. To the best of our knowledge, this represents **the first introduction of a variance-aware, column-wise shrinkage mechanism specifically tailored for cumulative-error compensation in LLM quantization**.

---

> > ### Author Response · Authors · 2025-11-21
> > **Rebuttal for Reviewer vGbd. Part [3]**
> >
> > ### **Empirical evidence: SOTA performance in extreme W2A4 and 2-bit weight-only regimes**
> >
> > We support the above conceptual and technical contributions with **system-level experiments**:
> > - In the **W2A4 setting**, we compare against representative SQ-based methods (SmoothQuant, OmniQuant, QuaRot, SpinQuant, OSTQuant, etc.) and obtain **SOTA performance** on WikiText2 perplexity and multiple zero-shot benchmarks (**Table 1**).
> > - In the **2-bit weight-only setting**, we directly compare with recent VQ-based SOTA (QuIP, QuIP#, PCDVQ, VPTQ, and AQLM in our updated experiments), achieving **SOTA** under the same benchmarks (**Table 4**).
> >
> > These results demonstrate that AEC-SVQ is not merely a simplistic combination of existing modules. Rather, it is a **cohesive framework designed under a unified objective that balances theoretical rigor with practical feasibility**.
> >
> > ### **Summary of innovations**
> >   - A **unified hybrid SQ–VQ framework** with a single learnable transform $T$ that simultaneously shapes weights, activations, and INT4 codebook structure under a joint W2A4 objective.
> >   - **CEAVQ**, which moves from local, single-layer proxies to a **cumulative-error-aware VQ** formulation with a closed-form optimal pre-compensated weight $W_{\text{opt}} = W + W G H^{-1}$ and a column-wise, feedback-aware VQ procedure.
> >   - A **column-wise adaptive shrinkage mechanism** for compensation derived from the same cumulative-error proxy, addressing bias–variance trade-offs **under limited calibration data.**
> >   - **Systematic SOTA results** in both W2A4 and 2-bit weight-only regimes, demonstrating that these innovations yield practical gains beyond a simple combination of existing SQ and VQ techniques.

---

> > > ### Author Response · Authors · 2025-11-21
> > > **Rebuttal for Reviewer vGbd. Part [4]**
> > >
> > > > How sensitive is your method to the choice of the calibration dataset?
> > >
> > > We thank the reviewer for raising this critical question. Calibration robustness is indeed vital for practical deployment. We tackle this challenge through **algorithmic design** and substantiate its effectiveness via **empirical verification**.
> > >
> > >
> > > ### **Methodological robustness to calibration data**
> > >
> > > - **Reliance on Low-Order Statistics**: The quantization and compensation processes depend solely on **first- and second-order statistics** of activations (e.g., $H = \mathbb{E}[XX^\top]$ and $G = \mathbb{E}[(\tilde{X} - X)X^\top]$). We do not rely on complex high-order features or task-specific labels, which makes our method **inherently less sensitive to the exact semantic composition of the calibration set**, as long as it reasonably excites typical activation patterns.
> > > - **Adaptive Bias-Variance Shrinkage**: Crucially, we address the sensitivity issue explicitly via our Adaptive Compensation mechanism (Section 3.3). We model the statistical noise arising from finite samples as a bias-variance trade-off3. By introducing a column-wise shrinkage factor $\lambda_k$ (derived in Eq. 9), **the algorithm automatically suppresses correction in directions with high estimation noise while retaining strength in reliable directions**. This acts as a data-driven regularizer that prevents the model from overfitting to specific artifacts in the calibration data.
> > >
> > > ### **Empirical Verification on sensitivity to calibration choice**
> > >
> > > Empirically, we have explicitly examined the sensitivity of AEC-SVQ to the calibration set, as summarized in **Table R2-1**:
> > > - Under the same model and quantization setting, we use **different calibration sets**.
> > >
> > > - As shown in **Table R2-1**, **both WikiText2 perplexity and zero-shot accuracy vary only within a relatively narrow band**.
> > >
> > > **Table R2-1: The impact of different calibration sets on the accuracy performance of AEC-SVQ.**
> > > | Model     | Method               | Bits | Wiki-PPL (↓) | ARC-c (↑) | ARC-e (↑) | BoolQ (↑) | HellaS. (↑) | OBQA (↑) |  PIQA (↑) | SIQA (↑) | WinoG. (↑) | Avg. (↑) |
> > > |:---------:|:--------------------:|:----:|:------------:|:---------:|:---------:|:---------:|:-----------:|:--------:|:---------:|:--------:|:----------:|:--------:|
> > > | llama3-8b | Full Precision       | W2A4 | 6.14         | 53.50     | 77.74     | 81.10     | 79.18       | 44.80    | 80.63     | 47.08    | 73.01      | 67.13    |
> > > |           | C4 calibrated        | W2A4 | 9.04         | 41.33     | 63.20     | 76.22     | 68.11       | 39.20    | 73.86     | 43.02    | 65.33      | 58.78    |
> > > |           | Redpajama calibrated | W2A4 | 9.23         | 42.12     | 64.58     | 75.24     | 70.14       | 27.40    | 75.11     | 42.75    | 65.01      | 57.79    |
> > > |           | Wikitext2 calibrated | W2A4 | 8.65         | 41.30     | 63.64     | 74.59     | 69.04       | 38.40    | 74.21     | 42.37    | 63.61      | 58.39    |
> > >
> > >
> > > These findings indicate that the performance of AEC-SVQ is attributed to its adaptive shrinkage mechanism. This design effectively **mitigates sensitivity to calibration data variations** and thereby increases the practical value of the approach.

---

> > > > ### Author Response · Authors · 2025-11-21
> > > > **Rebuttal for Reviewer vGbd. Part [5]**
> > > >
> > > > > Fine-tuning appears to have a significant impact on the results, which raises some concerns about the robustness of the proposed strategy. Could you elaborate on this point and confirm whether the comparisons with other works are conducted fairly (e.g., fine-tuning applied to your method but not to baselines)?
> > > >
> > > > We appreciate the reviewer's scrutiny regarding the impact of fine-tuning on robustness and fairness. We address this concern through three key clarifications:
> > > >
> > > > ### **1. This is PTQ-Compatible Lightweight Optimization, Not Task-Level Fine-Tuning**
> > > >
> > > > We would first like to clarify what we mean by “fine-tuning” in our work. We emphasize that our work strictly operates within the Post-Training Quantization (PTQ) paradigm. We do not perform large-scale, task-level fine-tuning of the model backbone:
> > > > - **Frozen Weights**: The main model weights ($W$) are never updated via gradients derived from downstream task losses.
> > > > - **Scope**: Following **standard practices in Vector Quantization** literature, we only perform lightweight optimization on a small set of quantization-specific parameters,using unlabeled calibration data. Specifically, **the VQ codebook vectors** and **LayerNorm affine parameters**.
> > > > - In this sense, our approach adheres to the same PTQ paradigm as methods like SpinQuant and OSTQuant. Consequently, there is no issue of unfair comparison.
> > > >
> > > > ### **2. Fairness of comparisons with baselines**
> > > >
> > > > The reviewer raised a concern regarding whether our method received preferential treatment via fine-tuning while baselines did not. We provide the following clarifications to demonstrate the fairness of our comparisons:
> > > >
> > > > - **In W2A4 Experiments (Comparison with SQ-based methods)**:
> > > >     - For AEC-SVQ, we only optimize the quantization-related parameters as described above.
> > > >     - For SQ-based baselines, we strictly follow the **official implementations and recommended calibration and parameter-optimization protocols,** including any post-training adaptation they support. We do not reduce their calibration data, number of optimization steps, or training budget.
> > > >     - Each method is therefore evaluated **under its best-practice PTQ configuration**, and we do not give AEC-SVQ any extra optimization.
> > > > - **In 2-bit Weight-Only Experiments (Comparison with VQ-based methods)**:
> > > >     - All VQ baselines are run with their **recommended settings**, including their own codebook optimization or light post-training updates when provided.
> > > >     - For AEC-SVQ, we use calibration data sizes, numbers of iterations, and optimization strategies that are of the same order of magnitude as those used for these baselines; we do not use larger datasets or significantly higher training budgets.
> > > >     - All comparisons were conducted under a fair premise where each method utilized its permitted PTQ and lightweight optimization protocols.
> > > >
> > > > ### **3. Influence of fine-tuning on robustness and relative performance**
> > > > Regarding the concern that "fine-tuning seems to have a significant impact on results," we offer two empirical observations to alleviate concerns about robustness:
> > > >
> > > > - **Superior Performance without Optimization**: As reported in the paper(**Table 1 and Table 3**), AEC-SVQ outperforms baseline methods even in a pure PTQ setting without any fine-tuning. This demonstrates that our proposed CEAVQ and adaptive compensation mechanisms possess strong intrinsic robustness and do not rely solely on subsequent parameter tuning to function effectively.
> > > > - **Progressive Improvement under VQ Conventions**: When we apply limited rounds of unsupervised optimization to the codebook and LayerNorm parameters (**standard practice in VQ literature**), the perplexity and zero-shot accuracy of AEC-SVQ show continuous but progressive improvements. Importantly, **this improvement does not alter the relative ranking of the methods.** We outperform existing baselines without fine-tuning, and the addition of lightweight optimization simply widens the performance gap in extremely low-bit settings.
> > > >
> > > > In summary, **our conclusions do not depend on a special or unfair fine-tuning procedure**. The proposed strategy remains within the **standard PTQ setting and standard practice in VQ**. All baselines are given their recommended implementations. The "fine-tuning" we apply is a standard, unsupervised quantization refinement step common to the VQ domain. **Even without it, AEC-SVQ establishes SOTA performance**, demonstrating that the robustness stems from the underlying Hybrid SQ-VQ framework and CEAVQ algorithm.

---

> > > > > ### Author Response · Authors · 2025-11-21
> > > > > **Rebuttal for Reviewer vGbd. Part [6]**
> > > > >
> > > > > > In Section A.5.2, is the analysis referring to OSTQ?
> > > > >
> > > > > We sincerely thank the reviewer for catching this oversight. We confirm that this is indeed a typo in the text of **Appendix A.5.2**.
> > > > >
> > > > > The analysis under "SPEEDUP AND MEMORY SAVINGS" (referencing **Table 9**) exclusively characterizes our proposed **AEC-SVQ W2A4 pipeline**. The erroneous mention of "OSTQuant" was a drafting error; OSTQuant serves only as a baseline for comparison in previous sections and was not the subject of this profiling analysis.
> > > > >
> > > > > We have corrected the misleading wording in the revised version to make it explicit that the speedup and memory analyses in **Appendix A.5.2** refer to AEC-SVQ W2A4, rather than to OSTQuant, and we will ensure this is clearly stated to avoid further ambiguity.

---

### Official Review · Reviewer_uQLA · 2025-10-31

**Soundness:** 2
**Presentation:** 3
**Contribution:** 2
**Rating:** 2
**Confidence:** 4

**Summary:**

The paper presents an approach for W2A4 (2-bit weight and 4-bit activation) quantization of large language models. While the motivation is clear and the experiments are conducted on relevant benchmarks, the contribution lacks sufficient novelty and fails to situate itself properly within the current literature on low-bit quantization.

**Strengths:**

1. The presentation is clear.
2. Extensive experiments are done on Llama and Qwen models.

**Weaknesses:**

1. The proposed method combines vector quantization for weights and scalar quantization for activations. THis is a setup that has already become standard practice in recent quantization research.
2. Without a clear conceptual or technical innovation, the contribution falls short of the standards for publication.
3. Several recent vector quantization approaches such as AQLM and QuIP# are not included in the comparison. These methods represent the state-of-the-art in efficient LLM quantization with vector quantization and should be considered essential baselines. The absence of such comparisons makes it difficult to assess the actual competitiveness of the proposed method. As it stands, the reported results cannot convincingly demonstrate superiority or even parity with existing solutions.

**Questions:**

1. It would strengthen the work if the authors provided hardware-level latency or energy efficiency evaluations, as quantization benefits are often hardware-dependent.

---

> ### Author Response · Authors · 2025-11-21
> **Rebuttal for Reviewer uQLA. Part [1]**
>
> **Dear Reviewer uQLA,**
>
> We sincerely thank you for your thorough evaluation and constructive comments. We address your valuable comments point by point as follows:
>
> ---
>
> > The proposed method combines vector quantization for weights and scalar quantization for activations. THis is a setup that has already become standard practice in recent quantization research.
>
>
> We thank the reviewer for carefully considering our method design.
> While we agree that **separately** using vector quantization (VQ) for weights and scalar quantization (SQ) for activations is closely related to existing quantization primitives, we would like to clarify that, to the best of our knowledge based on a survey of the last four years of relevant work, **there is no prior method that jointly employs weight VQ and activation SQ**.
> Our goal is not merely to adopt this high-level combination, but to develop **a unified, optimizable framework that jointly couples weight VQ, activation SQ, and INT4 codebook under the stringent W2A4 inference setting.** In this sense, we believe our method design goes beyond what is typically referred to as “standard practice”.
>
> While VQ and SQ are indeed common individual primitives, our contribution lies in the **systematic coupling of these techniques to unlock practical W2A4 inference (2-bit memory, 4-bit compute)**, which addresses a gap between two distinct research directions:
> - **Weight-only VQ line.** These works focus on aggressively compressing weights via VQ/PQ to approach 2-bit storage, but typically **keep activations in high precision** (e.g., FP16/BF16), and **rely on high-precision GEMM or non-trivial decoding operators.** As a result, they cannot directly realize practical low-bit integer GEMM kernels and thus do not provide deployable end-to-end integer inference.
> - **Weight–activation SQ line.** Methods such as SmoothQuant, SpinQuant, and OSTQuant primarily **reshape activation distributions to mitigate outliers, and apply scalar quantization to both weights and activations.** These approaches are mature in practice, but they do not exploit VQ on weights and therefore cannot fully leverage the storage and bandwidth advantages of VQ in the extreme W2A4 regime.

---

> > ### Author Response · Authors · 2025-11-21
> > **Rebuttal for Reviewer uQLA. Part [2]**
> >
> > Our approach extends **beyond a superficial high-level combination of weight VQ and activation SQ.** Instead, the core innovation lies in achieving structural and algorithmic synergy among these components under the strict constraints of W2A4 inference with INT4 codebooks.
> > From this unified perspective, **a naive combination of weight VQ and activation SQ inevitably suffers from severe cumulative errors and statistical noise.** To address these specific challenges, we propose a series of novel mechanisms designed to resolve these issues.
> > - **Learnable transform $T$ for joint shaping of weights, activations, and codebooks.**
> >   - For **weights**, the transform $T$ makes the transformed weight distribution more isotropic, thereby significantly reducing VQ reconstruction error.
> >   - For **activations**, under the same transform, the inverse-transformed activations have a contracted dynamic range that is more amenable to high-fidelity SQ.
> >   - On top of this, we design the **INT4 codebook** directly under INT4 compute constraints so that quantized weights map efficiently to INT4 GEMM kernels, rather than treating the codebook as an independent compression module that is decoupled from the integer operator.
> > - **CEAVQ (Cumulative-Error-Aware VQ).**
> > Traditional VQ for weights typically optimizes a local reconstruction error and does not explicitly account for **cross-layer cumulative error**, which becomes critical in the extreme W2A4 regime. In CEAVQ, we explicitly model both upstream quantization error and activation quantization error in a single proxy objective and derive a **closed-form pre-compensation term**. This ensures that each layer’s VQ step is optimized with respect to the downstream deviation from the full-precision output, rather than only minimizing local reconstruction error.
> > - **Adaptive bias–variance shrinkage for limited calibration data.**
> >   In realistic deployment scenarios, the calibration set is small and possibly distributionally shifted. Directly applying the full cumulative-error correction can overfit noisy statistics. We  formalize this as a **bias–variance trade-off** and derive **column-wise adaptive shrinkage coefficients** that retain strong compensation in reliably estimated directions, and automatically shrink compensation in high-variance directions. This design stabilizes CEAVQ under limited calibration data and reduces sensitivity to calibration noise.
> >
> > This unified design leads to concrete benefits in both accuracy and efficiency: as shown in **Table 1** in the initial submission, AEC-SVQ consistently outperforms strong PTQ baselines under the W2A4 setting across multiple LLaMA and Qwen families, and **Table 2 / Appendix A.5.2** report **2.2×–3.6× prefill speedup and up to 7.1× memory saving** compared to FP16.
> >
> > In summary, AEC-SVQ is **not** a simple combination of existing VQ and SQ techniques. It represents **a unified and learnable framework where weight VQ, activation SQ, and INT4 codebook quantization are co-designed under a rigorous shared objective.** By integrating **cumulative-error-aware compensation with variance-robust calibration**, we realize a practical and **high-accuracy W2A4 LLM inference** pipeline. This holistic strategy extends **beyond the standard isolated application of VQ or SQ and remains unexplored in current hybrid schemes.** We maintain that the systematic coupling and theoretical treatment of these components under stringent W2A4 constraints constitute a design paradigm that distinctively diverges from current standard practices.

---

> > > ### Author Response · Authors · 2025-11-21
> > > **Rebuttal for Reviewer uQLA. Part [3]**
> > >
> > > > Without a clear conceptual or technical innovation, the contribution falls short of the standards for publication.
> > >
> > > We respectfully argue that our work offers clear and significant innovations at both conceptual and technical levels. Below, we clarify our **core conceptual novelty** and the **three specific technical innovations** designed to achieve it.
> > >
> > > ### **1. Conceptual novelty: jointly optimizing 4-bit compute and 2-bit memory / bandwidth**
> > >
> > > Most existing LLM quantization methods predominantly optimize **either compute friendliness or storage efficiency**, but not both in a unified way:
> > > - **Compute-oriented line** (e.g., SmoothQuant, OmniQuant, QuaRot, SpinQuant, OSTQuant): these methods transform activations and apply scalar quantization so that models can run on INT8/INT4 kernels, but **the weight storage compression is limited**.
> > > - **Storage-oriented line** (e.g., QuIP, QuIP#, AQLM, PCDVQ, VPTQ): these methods use vector quantization to approach 2-bit weight storage, but are typically **weight-only**, and often **rely on high-precision operators or complex decoding**, making it difficult to directly deploy them on real INT4 GEMM kernels.
> > >
> > >
> > > **Our Conceptual Novelty**: We do not merely optimize either side in isolation. Instead, AEC-SVQ is the first framework designed to treat **4-bit Integer Computation** and **2-bit Memory Footprint** as co-equal optimization goals. This dual-objective perspective necessitates a fundamental redesign of how VQ and SQ interact, leading to the specific technical innovations detailed below.
> > > ### **2. Technical contributions**
> > > ###  **2.1 Technical contribution I: Unified hybrid SQ–VQ framework with a learnable linear transform**
> > >
> > > **Challenge.** SQ-based methods (SmoothQuant, SpinQuant, OSTQuant, etc.) mainly transform activation distributions to mitigate outliers and typically use scalar weight quantization; VQ-based methods (QuIP#, AQLM, PCDVQ, etc.) focus on compact weight coding and are often weight-only or tied to high-precision compute. These **two lines are largely designed in isolation**, and **there is no unified view that simultaneously incorporates weight VQ, activation SQ, and the structural constraints of integer operators.**
> > >
> > > **Our approach.**
> > > - On the theoretical side, we show (in Section 3 and the appendix) that there exists a linear transform of the form $T = \Lambda O$ that can **simultaneously tighten the upper bounds of the quantization errors for both weights and activations**, providing a formal basis for jointly reshaping their distributions.
> > > - On the algorithmic side, we explicitly introduce a **learnable linear transform** $T$ into the network to co-shape the joint distribution of weights and activations so that the transformed weights become more isotropic and better aligned with VQ, and the inverse-transformed activations have a contracted dynamic range that is more amenable to high-fidelity SQ.
> > > - When designing the weight codebook, we directly incorporate INT4 kernel constraints so that the quantized weights map naturally to INT4 GEMM, instead of treating VQ as a decoupled compression module.
> > >
> > > Thus, our framework is not a simple combination of existing VQ and SQ techniques: **it uses a single, learnable transform to jointly design weight VQ, activation SQ, and the INT4 codebook under a shared optimization objective**, enabling deployable W2A4 integer inference on large LLMs (see **Table 1 and Appendix A.5.2**).

---

> > > > ### Author Response · Authors · 2025-11-21
> > > > **Rebuttal for Reviewer uQLA. Part [4]**
> > > >
> > > > ### **2.2 Technical Innovation II: CEAVQ (Cumulative-Error-Aware Vector Quantization)**
> > > >
> > > > **Challenge.** Existing VQ-style methods (e.g., VPTQ, AQLM, GPTVQ) typically minimize local reconstruction errors, such as per-layer weight reconstruction loss on calibration data, or refine quantization via residual codebooks. This design **overlooks the accumulation of upstream quantization errors across layers**, which becomes particularly critical in the extreme W2A4 regime where both weights and activations are heavily compressed.
> > > >
> > > > **Our approach (CEAVQ).**
> > > > - We start from a **proxy loss** that explicitly accounts for both upstream and current-layer errors (Eq. (12)–(16)), and, via a Hessian approximation $H$, derive a **closed-form correction term** $W G H^{-1}$ that pre-compensates each layer’s weights before quantization.
> > > > - As a result, the objective of each layer is no longer simply minimize layer’s local quantization error, but instead **reduce the end-to-end output deviation induced by quantization**, which is especially important under W2A4 where activation errors also propagate.
> > > > - The full derivation is provided in **Appendix A.3**, and the ablation in **Table 3** shows that CEAVQ yields consistent improvements in perplexity and zero-shot accuracy across models.
> > > >
> > > > To our knowledge, explicitly modeling cumulative error in a VQ objective and deriving a **practical closed-form correction** of this form has not appeared in existing LLM quantization work, and we consider CEAVQ a clear technical innovation.
> > > >
> > > > ### **2.3 Technical contribution III: Adaptive bias–variance shrinkage for limited calibration data**
> > > >
> > > > **Challenge.** The theoretically derived correction depends on Hessian approximations and statistical estimates computed from **finite calibration data**. In practice, naively applying the theoretically optimal correction can **overfit the calibration set and degrade performance on real data**.
> > > >
> > > > **Our approach.**
> > > > - We view correction strength as a **ridge / shrinkage** problem and derive a **closed-form column-wise solution**, which can be written as an explicit shrinkage factor.
> > > > - We then design a rule that **adapts the shrinkage strength per column based on the estimated variance:** columns with higher noise receive stronger shrinkage, while low-noise columns retain a stronger correction. This yields a better bias–variance trade-off under realistic calibration budgets.
> > > >
> > > > To the best of our knowledge, this kind of **column-wise adaptive shrinkage mechanism specifically tailored to cumulative-error compensation in LLM quantization** has not been explored in prior work.
> > > >
> > > >
> > > > ### **3. Empirical evidence: SOTA performance in extreme W2A4 and 2-bit weight-only regimes**
> > > >
> > > > Our system-level experiments further support that AEC-SVQ is more than a simple stacking of existing modules:
> > > > - Under the **W2A4 setting**, we compare against strong SQ-based baselines such as SmoothQuant, OmniQuant, QuaRot, SpinQuant, and OSTQuant (**Table 1**). AEC-SVQ achieves **SOTA** results on WikiText2 perplexity and multiple zero-shot tasks.
> > > > - Under the **2-bit weight-only setting**, we compare against recent VQ-based methods such as QuIP, QuIP#, PCDVQ, and VPTQ (**Table 4**), and obtain **SOTA** performance under the same backbones and settings.
> > > >
> > > > These results indicate that AEC-SVQ is not a superficial combination of two existing lines, but a coherent framework designed around a unified objective, with theoretical backing and practical deployability on INT4 hardware.

---

> ### Author Response · Authors · 2025-11-21
> **Rebuttal for Reviewer uQLA. Part [5]**
>
> > Several recent vector quantization approaches such as AQLM and QuIP# are not included in the comparison. These methods represent the state-of-the-art in efficient LLM quantization with vector quantization and should be considered essential baselines. The absence of such comparisons makes it difficult to assess the actual competitiveness of the proposed method. As it stands, the reported results cannot convincingly demonstrate superiority or even parity with existing solutions.
>
> We thank the reviewer for this valuable suggestion. We agree that QuIP# and AQLM are essential baselines in the current landscape. We respectfully wish to clarify that we have indeed compared against SOTA VQ methods, and we have now added AQLM to ensure a comprehensive evaluation.
>
> **(1) Comparison in the 2-bit weight-only setting**
>
> First, we would like to clarify that in the **2-bit weight-only quantization** scenario, we already include a systematic comparison against several recent VQ-based SOTA methods(**QuIP, QuIP#, PCDVQ, and VPTQ**) in **Table 4** of the main paper. The reason AQLM was not reported in the initial submission is purely practical: our experiments are conducted on the more recent **LLaMA3-8B (released April 2024)**, while the **original AQLM paper (January 2024)** does not provide public results for this backbone, so we could not directly reuse or align with their reported numbers at the time of submission.
>
> To further address your concerns, we have now implemented and evaluated AQLM under the same 2-bit weight-only setting. The results are summarized in **Table R1-1** below.
>
> **Table R1-1: Performance of AEC-SVQ on 2bit weight-only quantization. AEC-SVQ outperforms methods designed specifically for this task, demonstrating its superior performance.**
>
> | model     | Method         | Bits | Wiki-PPL (↓) | ARC-c (↑) | ARC-e (↑) | HellaS. (↑) |  PIQA (↑) | WinoG. (↑) | Avg. (↑) |
> |:---------:|:--------------:|:----:|:------------:|:---------:|:---------:|:-----------:|:---------:|:----------:|:--------:|
> | llama3-8b | Full Precision | 16   | 6.14         | 53.50     | 77.74     | 79.18       | 80.63     | 73.01      | 72.81    |
> |           | GPTQ           | 2    | 210          | 19.90     | 28.80     | 27.70       | 53.00     | 50.50      | 36.16    |
> |           | DB-LLM         | 2    | 13.6         | 28.20     | 59.10     | 42.10       | 68.90     | 60.40      | 51.74    |
> |           | QuIP           | 2    | 85.1         | 21.30     | 29.00     | 29.20       | 52.90     | 51.70      | 36.81    |
> |           | **QuIP#**         | 2    | 9.11         | 39.20     | 72.90     | 51.60       | 75.60     | 68.20      | 61.50    |
> |           | **AQLM**           | 2.02 | 9.38         | 41.30     | 68.80     | 55.20       | 75.22     | 69.04      | 61.91    |
> |           | VPTQ           | 2.08 | 9.29         | 36.90     | 71.00     | 52.20       | 75.10     | 65.90      | 60.22    |
> |           | PCDVQ          | 2    | 8.77         | 37.54     | 71.75     | 51.57       | 74.59     | 67.56      | 60.60    |
> |           | **AEC-SVQ(ours)**  | 2    | **8.018**        | **40.10**     | **65.45**     | **71.05**       | **75.41**     | **68.03**      | **64.01**    |
>
>
> As shown in **Table R1-1**, **AEC-SVQ outperforms QuIP# and AQLM, as well as other VQ baselines, on both WikiText2 perplexity and average zero-shot accuracy.** This evidence confirms that AEC-SVQ extends beyond the W2A4 setting to rival or outperform state-of-the-art methods such as AQLM and QuIP# in the weight-only VQ regime for which they were specifically optimized.

---

> > ### Author Response · Authors · 2025-11-21
> > **Rebuttal for Reviewer uQLA. Part [6]**
> >
> > **(2) Comparison in the W2A4 setting**
> >
> > We also understand the implicit concern about **vector-quantization SOTA under the W2A4 setting**, which is the main focus of our work. It is important to note that QuIP#, AQLM, and related methods are originally proposed as high-compression, weight-only schemes, and their public implementations do not directly support W2A4 quantization or INT4 GEMM deployment.
> >
> > To still provide as fair a comparison as possible, we extended these methods to a W2A4 setting by keeping their original weight quantization pipeline intact, and equipping them with a 4-bit activation quantization procedure.
> >
> > **Table R1-2:Complete Comparison of the perplexity score on WikiText2 and averaged accuracy on Zero-shot Common Sense Reasoning tasks on LLaMA-3-8B**
> > | Model     | Method         | Bits | Wiki-PPL (↓) | ARC-c (↑) | ARC-e (↑) | BoolQ (↑) | HellaS. (↑) | OBQA (↑) |  PIQA (↑) | SIQA (↑) | WinoG. (↑) | Avg. (↑) |
> > |:---------:|:--------------:|:----:|:------------:|:---------:|:---------:|:---------:|:-----------:|:--------:|:---------:|:--------:|:----------:|:--------:|
> > | llama3-8b | Full Precision | W2A4 | 6.14         | 53.50     | 77.74     | 81.10     | 79.18       | 44.80    | 80.63     | 47.08    | 73.01      | 67.13    |
> > |           | AQLM           | W2A4 | 52577.52     | 23.46     | 27.35     | 48.64     | 26.30       | 28.80    | 51.72     | 34.17    | 50.22      | 36.33    |
> > |           | QuIP#          | W2A4 | 2834.46      | 24.03     | 30.28     | 52.48     | 27.91       | 25.80    | 50.41     | 33.02    | 51.97      | 36.99    |
> > |           | AEC-SVQ        | W2A4 | 8.65         | 41.30     | 63.64     | 74.59     | 69.04       | 38.40    | 74.21     | 42.37    | 63.61      | 58.39    |
> >
> >
> > As shown in **Table R1-2**, existing VQ SOTA methods suffer from substantial degradation once 4-bit activations are introduced: **the lack of explicit handling of activation outliers and cumulative quantization error leads to significantly worse perplexity and zero-shot accuracy**, sometimes approaching chance-level performance on more challenging tasks.
> >
> > In contrast, **AEC-SVQ is explicitly designed to co-optimize weight VQ, activation SQ, and INT4 codebook quantization**, and in the same W2A4 setting it clearly outperforms both these VQ methods and strong SQ baselines on most zero-shot benchmarks,  while simultaneously achieving 2-bit weight storage and deployable INT4 GEMM inference.
> >
> > **(3) Summary**
> >
> > We hope to convey two key messages:
> > - In the **2-bit weight-only VQ setting**, we now include direct comparisons to QuIP#, AQLM, and other recent VQ methods, and **AEC-SVQ achieves SOTA performance** under the same evaluation.
> > - In the more practically relevant **W2A4 setting**, existing VQ SOTA methods are not robust once 4-bit activations are introduced, whereas **AEC-SVQ remains accurate** and efficient, which we view as a central contribution of this work.

---

> > > ### Author Response · Authors · 2025-11-21
> > > **Rebuttal for Reviewer uQLA. Part [7]**
> > >
> > > > It would strengthen the work if the authors provided hardware-level latency or energy efficiency evaluations, as quantization benefits are often hardware-dependent.
> > >
> > > We fully agree with the reviewer that the practical value of any quantization method hinges on its deployability and efficiency on real hardware.
> > > In fact, the current version of the paper already includes **hardware-level latency and memory evaluations**. Specifically, in **Table 2 of the main paper and Table 9 in Appendix A.5.2**, we report:
> > > - **Prefill latency** of W2A4 (AEC-SVQ) versus FP16
> > > - **GPU memory consumption**
> > >
> > > on an Intel Xeon Gold 5317 CPU + NVIDIA RTX 3090 GPU platform, across multiple LLaMA model sizes and sequence lengths. These measurements directly reflect real-device inference behavior. For example, on a 30B-scale model with long sequences, **AEC-SVQ achieves over 3× prefill speedup and around 7× memory reduction** compared to the FP16 baseline (see **Appendix Table 9**), demonstrating that the proposed W2A4 design translates into substantial end-to-end gains, not just theoretical compression.
> > >
> > > These results directly demonstrate that AEC-SVQ translates its theoretical compression efficiency into tangible reductions in latency and memory footprint on commodity GPUs. We will ensure these findings are referenced more prominently in the revised version.

---

> ### Author Response · Authors · 2025-11-28
> **Gentle Follow-up in the Final Week of Discussion**
>
> **Dear Reviewer uQLA,**
>
> Thank you again for your time and the constructive comments provided during the review process. As the discussion period is entering its final stage, we would like to kindly follow up to ensure that our response and new experimental results have adequately addressed your concerns.
>
> Following your suggestions, we have updated our manuscript and rebuttal to include the following:
>
> - **Clarified the Conceptual and Technical Novelty:** We provided a **detailed breakdown distinguishing AEC-SVQ from a “standard practice” of VQ and SQ.** We highlighted our unified, learnable framework that integrates the **joint linear transform, Cumulative-Error-Aware VQ (CEAVQ), and adaptive bias–variance shrinkage.** This design allows us to co-optimize for 4-bit compute and 2-bit memory, achieving a **synergy not explored in prior isolated VQ or SQ works**.
> - **Added Comparisons with SOTA VQ Baselines (AQLM & QuIP#):** We conducted additional experiments comparing AEC-SVQ against AQLM and QuIP#. The results (**Table R1-1** and **Table R1-2**) demonstrate that **our method outperforms these strong baselines in the 2-bit weight-only setting and maintains robustness in the W2A4 setting,** where traditional VQ methods suffer significant degradation due to activation quantization errors.
> - **Highlighted Hardware Efficiency Evaluations:** We directed attention to the hardware performance data (latency and memory footprint) included in **Table 2** and **Appendix A.5.2**. These results confirm that our theoretical efficiency translates into practical gains, **achieving 2.2×–3.6× prefill speedup and up to 7.1× memory savings on commodity GPUs compared to FP16.**
>
> We completely understand that you may have many commitments, and we are truly grateful for any time you could spare. We remain fully available to answer any further questions you may have. We respectfully hope you might reconsider the evaluation of our work.
>
> Thank you again for your thoughtful evaluation and valuable insights.
>
> Sincerely,
>
> The Authors

---

### Meta-Review · Area_Chair_8RuX · 2026-01-07

**Summary:**

This paper focuses on a specific quantization setting of W2A4 and introduces a hybrid SQ-VQ framework. Besides technical details, all the reviewers show concerns about the novelty, even for the only reviewer who gave a positive rating. The authors spent great effort in providing a lengthy rebuttal. Unfortunately, this AC can’t see a clear and concise articulation of the novelty at the high level. Swamping readers with lengthy details does not help. The paper needs a major rewrite to articulate its novelty more concisely and intuitively.

**Reviewer Concerns:**

See above.

**Reviewer Scores:**

No. The rebuttal is too long and hard to get the key points at the high level.

---

### Decision · Program_Chairs · 2026-01-26

Reject